# The brown adipocyte protein CIDEA promotes lipid droplet fusion via a phosphatidic acid-binding amphipathic helix

David Barneda[1], Joan Planas-Iglesias[2], Maria L Gaspar[3], Dariush Mohammadyani[4], Sunil Prasannan[5], Dirk Dormann[6], Gil-Soo Han[7], Stephen A Jesch[3], George M Carman[7], Valerian Kagan[4], Malcolm G Parker[1], Nicholas T Ktistakis[8], Judith Klein-Seetharaman[2,4], Ann M Dixon[5], Susan A Henry[3], Mark Christian[1,2]*

[1]Institute of Reproductive and Developmental Biology, Imperial College London, London, United Kingdom; [2]Warwick Medical School, University of Warwick, Coventry, United Kingdom; [3]Department of Molecular Biology and Genetics, Cornell University, Ithaca, United States; [4]Department of Bioengineering, University of Pittsburgh, Pittsburgh, United States; [5]Department of Chemistry, University of Warwick, Coventry, United Kingdom; [6]Microscopy Facility, MRC Clinical Sciences Centre, Imperial College London, London, United Kingdom; [7]Department of Food Science, Rutgers Center for Lipid Research, Rutgers University, New Brunswick, United States; [8]Signalling Programme, Babraham Institute, Cambridge, United Kingdom

**Abstract** Maintenance of energy homeostasis depends on the highly regulated storage and release of triacylglycerol primarily in adipose tissue, and excessive storage is a feature of common metabolic disorders. CIDEA is a lipid droplet (LD)-protein enriched in brown adipocytes promoting the enlargement of LDs, which are dynamic, ubiquitous organelles specialized for storing neutral lipids. We demonstrate an essential role in this process for an amphipathic helix in CIDEA, which facilitates embedding in the LD phospholipid monolayer and binds phosphatidic acid (PA). LD pairs are docked by CIDEA trans-complexes through contributions of the N-terminal domain and a C-terminal dimerization region. These complexes, enriched at the LD–LD contact site, interact with the cone-shaped phospholipid PA and likely increase phospholipid barrier permeability, promoting LD fusion by transference of lipids. This physiological process is essential in adipocyte differentiation as well as serving to facilitate the tight coupling of lipolysis and lipogenesis in activated brown fat.

*For correspondence: m.christian@warwick.ac.uk

Competing interests: The authors declare that no competing interests exist.

## Introduction

Evolutionary pressures for survival in fluctuating environments that expose organisms to times of both feast and famine have selected for the ability to efficiently store and release energy in the form of triacylglycerol (TAG). However, excessive or defective lipid storage is a key feature of common diseases such as diabetes, atherosclerosis, and the metabolic syndrome (*Greenberg et al., 2011*). The organelles that are essential for storing and mobilizing intracellular fat are lipid droplets (LDs) (*Walther and Farese, 2012*). They constitute a unique cellular structure where a core of neutral lipids is stabilized in the hydrophilic cytosol by a phospholipid monolayer embedding LD proteins. While most mammalian cells present small LDs (<1 μm) (*Suzuki et al., 2011*), white (unilocular)

**eLife digest** If other energy sources become unavailable, cells fall back on stores of fatty molecules called lipids. These are held in membrane-enclosed compartments in the cell called lipid droplets, which in mammals are particularly abundant in fat cells called adipocytes. There are two main types of adipocytes: white adipocytes have a single giant lipid droplet, whereas brown adipocytes contain many smaller droplets.

Proteins embedded in the membrane that surrounds a lipid droplet help to control the droplet's growth and when it releases lipids. For example, a protein called CIDEA, which is only found in brown adipocytes, helps lipid droplets to grow by enabling one droplet to transfer its contents to another droplet. However, little is known about how this occurs.

By combining cell biology, biophysical and computer modelling approaches, Barneda et al. investigated how normal and mutant forms of CIDEA affect the growth of lipid droplets. These experiments identified a helix in the structure of CIDEA that embeds it in the membrane, from where it can then interact with CIDEA proteins on other lipid droplets to hold the droplets together. In addition, the helix interacts with a molecule in the lipid droplet membrane called phosphatidic acid. Barneda et al. suggest that this interaction helps to transfer the contents of one droplet to another by making it easier for lipids to move through the droplets' membranes.

The next challenge is to characterize the mechanisms that control CIDEA activity to influence the formation of the multiple lipid droplets that distinguish brown and BRITE (brown-in-white) adipocytes from white adipocytes. The lipid droplets in brown adipocytes are an important target for research to combat obesity, due to the 'burning' rather than storing of lipids that occurs in these cells.

adipocytes contain a single giant LD that occupies most of their cell volume. In contrast, brown (multilocular) adipocytes hold multiple LDs of smaller size that increase the LD surface/volume ratio, which facilitates the rapid consumption of lipids for adaptive thermogenesis (*Cinti, 2012*).

The exploration of new approaches for the treatment of metabolic disorders has been stimulated by the rediscovery of active brown adipose tissue (BAT) in adult humans (*Virtanen et al., 2009*; *Cypess and Kahn, 2010*) and by the induction of multilocular brown-like cells in white adipose tissue (WAT) (*Harms and Seale, 2013*). The multilocular morphology of brown adipocytes is a defining characteristic of these cells along with expression of genes such as Ucp1. The acquisition of a unilocular or multilocular phenotype is likely to be controlled by the regulation of LD growth. Two related proteins, CIDEA and CIDEC, promote LD enlargement in adipocytes (*Wu et al., 2014*; *Puri et al., 2007*; *2008*), with CIDEA being specifically found in BAT. Together with CIDEB, they form the CIDE (cell death-inducing DFF45-like effector) family of LD proteins, which have emerged as important metabolic regulators (*Xu et al., 2012*).

Different mechanisms have been proposed for LD enlargement, including in situ neutral lipid synthesis, lipid uptake, and LD–LD coalescence (*Kuerschner et al., 2008*; *Wilfling et al., 2013*; *Boström et al., 2007*). The study of CIDE proteins has revealed a critical role in the LD fusion process in which a donor LD progressively transfers its content to an acceptor LD until it is completely absorbed (*Gong et al., 2011*). However, the underlying mechanism by which CIDEC and CIDEA facilitate the interchange of TAG molecules between LDs is not understood. In the present study, we have obtained a detailed picture of the different steps driving this LD enlargement process, which involves the stabilization of LD pairs, phospholipid binding, and the permeabilization of the LD monolayer to allow the transference of lipids.

## Results

### CIDEA expression mimics the LD dynamics observed during the differentiation of brown adipocytes

To examine the processes controlling LD enlargement in brown adipocytes, we followed LD dynamics using time-lapse microscopy. During differentiation of immortalized brown pre-adipocytes, large

LDs were formed by the fusion of pre-existing LDs (*Video 1*). This fusion process was characterized by a slow and progressive reduction in the volume of a donor LD until it was completely absorbed by an acceptor LD (*Figure 1A*), which is characteristic of CIDE activity. As CIDEA is selectively expressed in brown adipocytes and could have a prominent role in the acquisition of their multilocular morphology, we explored the effects of its expression in undifferentiated pre-adipocytes. After inducing CIDEA, LDs in pre-adipocytes showed an equivalent dynamic pattern to that observed in differentiating brown cells, with the progressive fusion of the initial LDs until a few large LDs remained in the cell (*Video 2*). LD fusion was achieved by the slow transference of lipids between LDs, and was preceded by the formation of small clusters of interacting LDs (*Figure 1B*). Given the importance of this process in adipocyte dynamics, we decided to undertake a comprehensive molecular analysis.

## Phases of CIDEA activity: LD targeting, LD–LD docking, and LD growth

The ectopic expression of full-length CIDEA induced the formation of large LDs through LD fusion by lipid transfer (*Figure 1C,D*). Control cells, which lacked expression of CIDEA, did not show LD enlargement (*Figure 1C*). As many proteins are constructed of domains, which are conserved across families and serve as their main structural and functional units, we assessed the conserved regions within the CIDE proteins. By comparing the 217 amino acid (aa) sequence of CIDEA with that of CIDEB and CIDEC, four highly conserved regions could be identified (*Figure 2A* and *Figure 2—figure supplement 1*). The N-terminal (N-term) domain of CIDEA is composed of a basic region (2–72 aa) followed by an acidic sequence (73–110 aa). These distinctly differently charged regions are indicated by protein crystallography studies to be important for the dimerization of CIDE domain proteins (*Lugovskoy et al., 1999*; *Wang et al., 2012*; *Sun et al., 2013*; *Lee et al., 2013*). The CIDEA C-terminal (C-term) is rich in basic aas and contains a highly conserved region (126–155 aa) and a basic and hydrophobic sequence (162–197 aa). Based on this sequence analysis, we created an extensive collection of v5-tagged CIDE point and deletion mutants to test their effects on LDs (*Figure 2*). Interestingly, certain mutations such as R171E/R175E promoted the aggregation of the cellular LDs in a few 'bunch of grapes'-like LD clusters, but were unable to induce the transference of lipids between them (*Figure 2B*). In other cases, as with the expression of CIDEA-(116–217)-v5, the LDs remained small and dispersed throughout the cytoplasm despite the protein being normally localized at their surface. Finally, some versions of CIDEA, such as CIDEA-(1–118)-v5, showed no LD localization and did not affect their size, number, or distribution. Together with the time-lapse results, this indicates that the molecular mechanism of CIDEA is composed of three discrete phases: LD targeting, LD–LD docking, and LD growth.

## A cationic amphipathic helix in C-term drives LD targeting

All the CIDEA constructs that showed impaired LD localization contained deletions or mutations in the C-term hydrophobic and basic region (162–197 aa) (*Figure 2C*). In fact, the last 66 aas of CIDEA were sufficient for LD localization, as shown with the expression of CIDEA-(152–217)-v5, whereas it lacked the ability to facilitate the docking of LDs (*Figures 2C* and *4A*). Although it is known that the C-term domain of CIDE proteins is essential for LD localization and enlargement (*Liu et al., 2009*; *Christianson et al., 2010*), only the structure of the N-term domain has been solved (*Lugovskoy et al., 1999*; *Wang et al., 2012*; *Sun et al., 2013*; *Lee et al., 2013*). The CIDE-N domain (Pfam reference

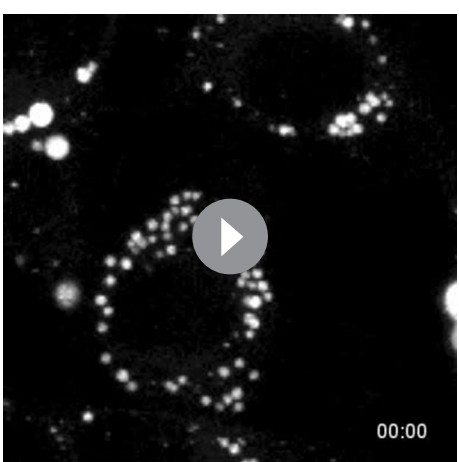

**Video 1.** Lipid droplet (LD) enlargement in differentiating immortalized brown adipose tissue (imBAT) cells. Immortalized brown pre-adipocytes were induced to differentiate by incubation for 48 hr + 6 hr with the described differentiation cocktails. The cell displays the characteristic LD enlargement pattern triggered by CIDE proteins, defined by the progressive fusion by lipid transference of the pre-existing LDs.

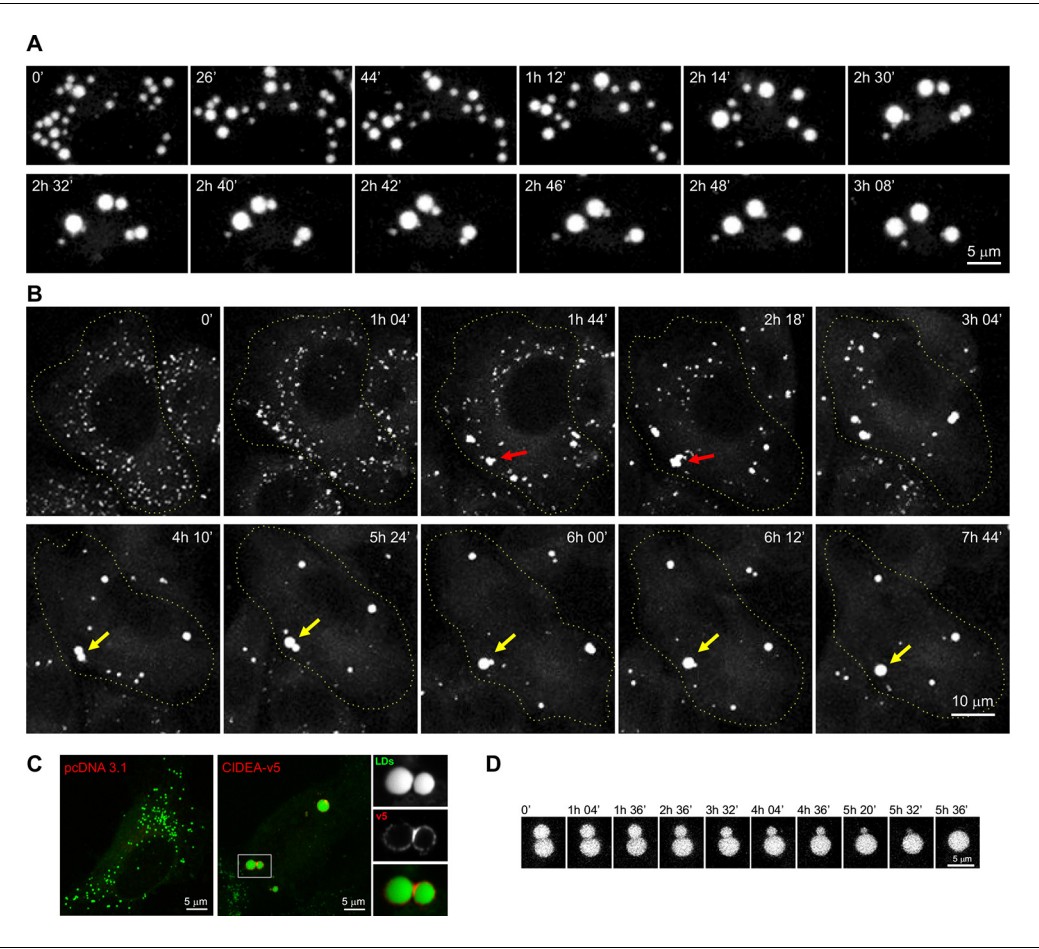

**Figure 1.** CIDEA promotes lipid droplet (LD) enlargement by transference of lipids. (**A**) Live imaging of the LD dynamics during the differentiation of a brown pre-adipocyte, showing the characteristic CIDE-triggered LD enlargement, characterized by the progressive transference of lipids from a donor to an acceptor LD until it is completely absorbed. (**B**) Live imaging of the LD dynamics in an undifferentiated 3T3-L1 cell 6 hr after infection with adenoviral particles carrying the Cidea gene. Red arrows highlight the transient formation of irregularly shaped LD clusters, while yellow arrows mark the fusion of two droplets by transference of lipids. (**C**) CIDEA-v5 expression in Hela cells induces LD enlargement. An enrichment in CIDEA-v5 (red) can be observed in the contact site between two LDs (green). (**D**) Detail of LD fusion by slow transference of lipids in a 3T3-L1 cell stably expressing CIDEA-v5.

PF02017) has been determined in members of the CIDE family (PBD Codes: 2eel (hCIDEA), 1D4B (hCIDEB), 4MAC (mCIDEC), 4ikg (mCIDEC)) to aas 40–117, 34–100, and 41–118 in hCIDEA, hCIDEB, and mCIDEC, respectively. Thus, the sequence 163–180 that we found essential for LD targeting (*Figure 2C*) lacks direct structural information to date. We therefore predicted its structure using in silico approaches. The region displayed high probability of a helical conformation with a strongly amphipathic character (*Figure 3A,B*, and *Figure 3—figure supplement 1*). To experimentally confirm the presence of an amphipathic helix in the LD-targeting domain of CIDEA, circular dichroism (CD) spectroscopy was used to estimate the secondary structure of a synthetic peptide corresponding to residues 158–185 in CIDEA. The CD spectra in the presence of 0.1% n-dodecyl-β-D-maltopyranoside confirmed the presence of α-helical structure (*Figure 3C*).

As some LD proteins are known to be bound to the LD membrane through amphipathic helices (*Hinson and Cresswell, 2009*; *Krahmer et al., 2011*), we tested if this short sequence was sufficient for LD targeting. While HA-(1–120)-CIDEA showed no LD localization and had no effect on LD distribution or size, HA-CIDEA-(1–117)-(163–180) was partially localized on the LD surface and promoted LD clustering (*Figure 3D*). Furthermore, the deletion of this C-term sequence in CIDEA-△(163–179)-

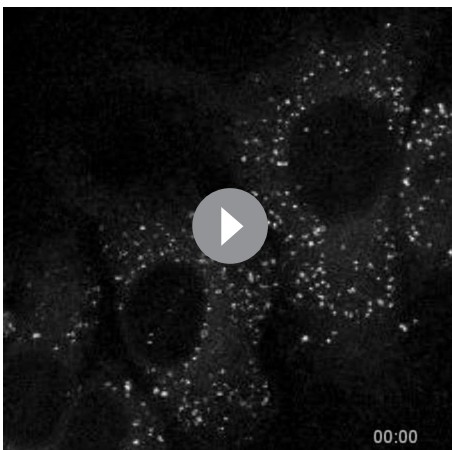

**Video 2.** Lipid droplet (LD) enlargement induced by CIDEA. LD dynamics in undifferentiated 3T3-L1 cells 6 hr after infection with adenoviral particles carrying the mouse Cidea gene. After CIDEA induction, the initial individual LDs form stable contacts reflected by small irregularly shaped clusters of LDs. These interacting LDs undergo an enlargement process by lipid transference, characterized by the progressive enlargement of the acceptor LD and shrinkage of the donor LD until only a few large LDs remain in the cell.

v5 completely eliminated LD localization in most of the cells (*Figure 2B,C*), confirming its role in LD targeting. However, partial LD localization could be observed in a small percentage of cells, together with the presence of LD clusters. This was also observed in CIDEA-△(162–197)-v5 and was particularly frequent in CIDEA-(N172X)-v5, which contains a deletion in the middle of the helix. In contrast, no LD targeting could be observed for the N-term fragment alone (1–118 aa) (*Figure 2C*). This may indicate that other regions in C-term may contribute to LD localization either by directly binding the LD membrane or by interacting with other LD proteins. Similarly, LD targeting was compromised when the amphipathic character of the helix was disrupted in CIDEA-(F166R/V169R/L170R)-v5 by introducing cationic aas in its hydrophobic face. Although LD localization was only lost in a small percentage of cells, in the remaining cells the LD staining was accompanied by a predominantly cytosolic localization (*Figures 2C* and *3E*). In contrast, the predominant LD localization of wild type (wt) CIDEA was maintained in CIDEA-(K167E/R171E/R175E)-v5, which presents a charge inversion of the helix but maintains its amphipathic properties (*Figures 2C* and *3E*).

## The amphipathic helix is essential for LD enlargement

In addition to its role in LD targeting, our data indicate that the cationic amphipathic helix in the C-term participates in the TAG transference step of CIDEA activity, as the charge inversion (K167E/R171E/R175E) did not affect LD targeting but completely blocked LD enlargement (*Figure 2C*). Despite not being essential for LD targeting, the cationic aas in the helix are highly conserved in vertebrates *Figure 3—figure supplement 1A*. K167 is 100% conserved across all vertebrate species examined. R171 was conserved across vertebrates including birds, snakes, lizards, crocodiles, turtles, marsupials, placental mammals, and monotremes, although not in fish. R175 is also highly conserved, with only birds, dolphins, and the Nile Tilapia (a fish) lacking this residue. Remarkably, an amphipathic helix is predicted in CIDEA of all the vertebrate species examined (*Figure 3—figure supplement 1B*).

The absence of negative charges in the helix appeared to be an essential condition to permit TAG transference, as a single inverted charge mutation such as R171E or R175E was sufficient to block LD enlargement (*Figure 2C*). In contrast, conservative substitutions such as R171K or K167R did not affect CIDEA activity, and even the substitution of the three basic aas with histidine in (K167H/R171H/R175H)-CIDEA-v5 was compatible with the formation of large LDs, strongly supporting the conclusion from sequence comparison that positive charges are required at these positions. As histidine has a lower pKa value than arginine and lysine, it can carry a positive charge depending on the pH and local environment, which could explain the activity retained by this protein.

## LD–LD docking is induced by the formation of CIDEA complexes

Deletions in the N-term domain of CIDEA impaired LD–LD docking, as shown by the increase in cells displaying isolated LDs (*Figure 2C*). Furthermore, LD clustering could be induced by forcing the LD localization of the N-term fragment through conjugation with the 18-aa amphipathic helix (HA-CIDEA-(1–117)-(162–180)) (*Figure 3C*).

As the N-term of CIDEA forms a highly polarized structure that is prone to dimerize (*Lugovskoy et al., 1999*), we hypothesized that LD–LD docking was induced by the N-term–N-term

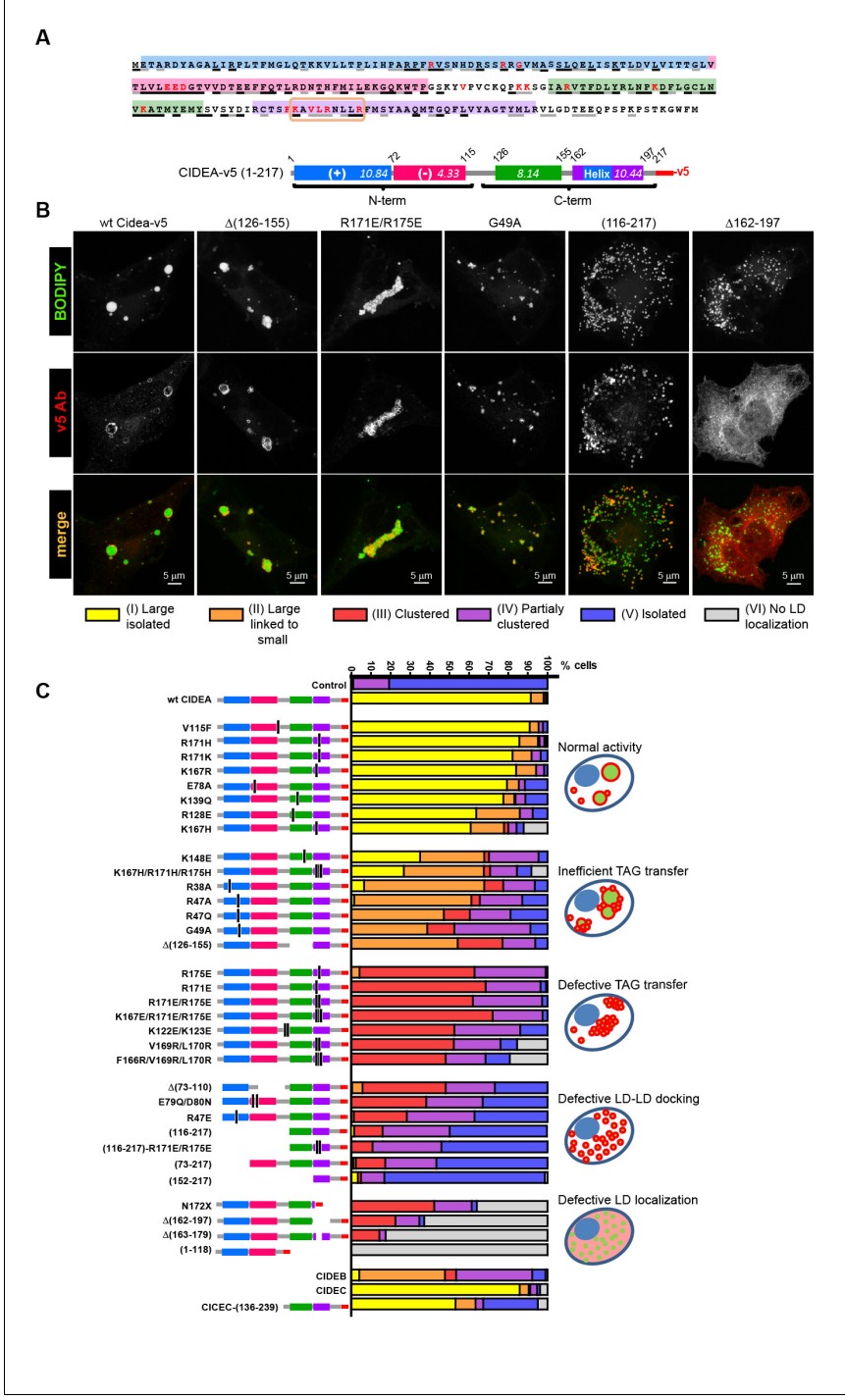

**Figure 2.** Mapping the functional domains of CIDEA. (**A**) Amino acid (aa) sequence of murine CIDEA highlighting the residues conserved in either CIDEB or CIDEC (grey underline), or in both proteins (black underline). The substituted aa in mutant constructs appear in red, and a positively charged sequence necessary for the TAG transfer step is encircled in orange. Four highly conserved regions are defined and symbolized by colour boxes in a linear representation of CIDEA-v5. The theoretical isoelectric point of each fragment is indicated inside the boxes. (**B**) Representative images of the different phenotypes observed in Hela cells overexpressing mutated forms of CIDEA-v5 24 hr after transfection cells were treated with oleic acid and incubated for further 18 hr prior to fixation. Cells were classified into six major phenotypes. Cells expressing fully active forms of CIDEA had few and large LDs (Type I). In some mutants, the large LDs remained attached to many small LDs, indicating that lipid transfer was inefficient or inactive for some LDs (Type II). When CIDEA alterations blocked the lipid transfer process, the LDs remained small and grouped in a few large clusters (Type III). If this was accompanied by inefficient LD–LD docking, the cells contained a number of small LD clusters combined with isolated LDs (Type IV). The CIDEA forms that could not stabilize LD–LD interactions displayed a phenotype similar to the mock transfected cells, with most of the LDs dispersed through

*Figure 2 continued on next page*

Figure 2 continued

the cytoplasm (Type V). Finally, some CIDEA constructs were unable to target the LDs, indicating an alteration of the LD-binding domain (Type VI). (**C**) Morphologic distribution of cells expressing each of the studied CIDE constructs. The phenotypic distribution was performed in a minimum of three independent experiments for every construct (n>50 cells).

The following figure supplement is available for figure 2:

**Figure supplement 1.** Alignment of amino acid sequences of CIDEA, CIDEB, and CIDEC.

interaction of CIDEA molecules in adjacent LDs (trans complexes). However, the C-term fragment (116–217) retained some degree of LD–LD docking activity (*Figure 2C*), indicating that an additional interaction site could be present in this region. In fact, a complete blocking of LD clustering was only observed with the shorter fragment 152–217, which lacks the N-term and a section of the C-term (*Figure 4A*).

The formation of CIDEA–CIDEA complexes was confirmed by co-immunoprecipitation (co-IP) of CIDEA-v5 with CIDEA-HA. Surprisingly, co-IP was observed with CIDEA-(116–217)-v5 but not CIDEA-(1–118)-v5, indicating that the C-term was responsible for that interaction (*Figure 4B*). Interestingly, a similar percentage of the input was co-immunoprecipitated for constructs producing highly clustered LDs (CIDEA-(R171E/R175E)-v5) and constructs showing few LD–LD contacts (CIDEA-v5 or CIDEA-(116–217)-v5). Hence, this C-term interaction is largely independent of the presence of LD–LD contacts, indicating that it may also occur in cis.

Within the C-term region, the deletion of the 162–197 sequence did not affect the co-IP whereas the signal was largely reduced in CIDEA-△(126–155)-v5 (*Figure 4B*), indicating that this conserved region was involved in the C-term interaction. However, the residual interaction still detectable by co-IP could sustain the LD-docking activity, as cells expressing this construct displayed normal LD clustering (*Figure 4A,B*). CIDEA-(152–217)-v5 (*Figures 2* and *4A*), which showed no LD clustering and lacked both the 126–155 interaction site and the N-term domain, displayed a further reduction on the co-IP signal (*Figure 4B*). Therefore, trans complexes through N-term dimerization would be responsible for the LD clusters and weak co-IP signal observed in CIDEA-(126–155)-v5. The lack of co-IP between the N-term fragment and the full-length CIDEA could be due to conformational and positional factors favouring the interaction between the HA-tagged full-length proteins in the LD or between the cytosolic v5-tagged N-term fragments. In fact, co-IP between N-term fragments of CIDEC was previously reported (*Sun et al., 2013*). This interaction could be disrupted with the point mutations E87Q/D88N or R55E as predicted by the crystal structure of the N-term fragment, which reveals the formation of homodimers in which the positively charged R46, R55, and R56 in one molecule interact with negative residues in the other (E87 and D88) (*Sun et al., 2013*). Interestingly, we found that the equivalent mutations in CIDEA (E79Q/D80N and R47E) impaired LD docking, while R47Q and R47A, which would not create repulsions between the interacting domains, did not affect CIDEA activity (*Figure 2C*). Taken together, these results suggest that both the C-term dimerization site (126–155) and the N-term domain of CIDEA can contribute to LD–LD docking by forming complexes with its counterparts on the adjacent LD.

## CIDEC differs from CIDEA in its dependence on the N-term domain

The differential expression of CIDEA and CIDEC in BAT and WAT could be related to the acquisition of multilocular or unilocular morphologies in brown and white adipocytes (*Barneda et al., 2013*). While the ectopic expression of both CIDEA and CIDEC produce LD enlargement in a similar manner, specific differences in their activity and regulation could achieve discrete outcomes. In fact, whereas deletion of the N-term domain of CIDEA blocks LD enlargement (*Figure 2C*), it has been described that the C-term fragment of CIDEC retains its activity (*Gong et al., 2011*; *Jambunathan et al., 2011*). Here we show that similar to CIDEA, the N-term of CIDEC is involved in LD–LD docking, as its deletion increases the fraction of cells displaying isolated LDs (*Figure 2C*). However, in the cells where the C-term of CIDEC could effectively induce LD–LD docking, large LDs were observed instead of LD clusters, showing that although its docking efficiency is reduced, this region of CIDEC is sufficient for docking and enlarging the LDs. This differs from CIDEA, in which the C-term fragment cannot induce LD enlargement despite retaining a partial LD docking activity.

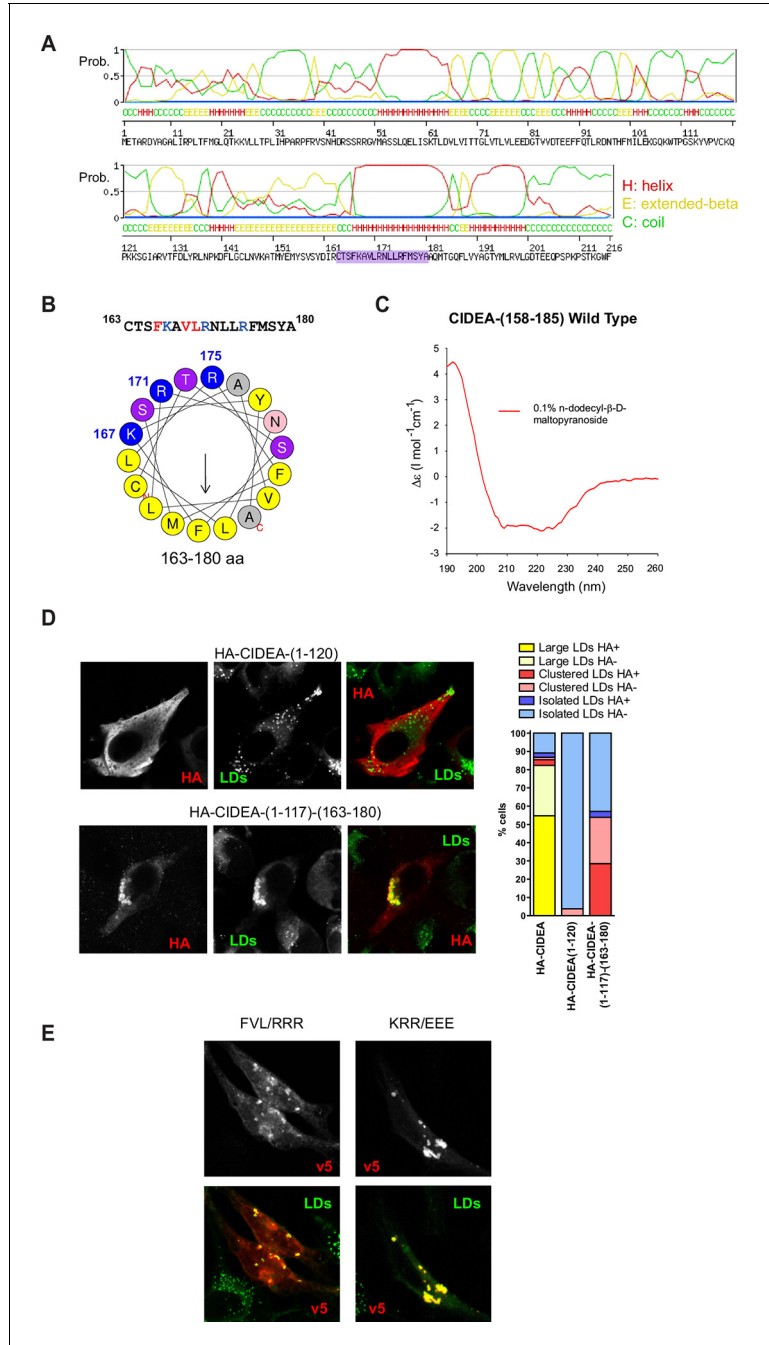

**Figure 3.** CIDEA targets the LD monolayer through a cationic amphipathic helix. (**A**) Secondary structure of CIDEA predicted by SWISS-MODEL server. (**B**) Helical wheel representation of the putative amphipathic α-helix (163–180) generated at http://heliquest.ipmc.cnrs.fr/. (**C**) Circular dichroism (CD) spectra of a 28-aa peptide corresponding to the 158–185 sequence of CIDEA (41 µM) solubilized in 50 mM potassium phosphate, pH 6.2 plus 0.1% n-dodecyl-β-D-maltopyranoside. (**D**) A Hela cell expressing HA-CIDEA-(1–120)-v5 (red) or HA-CIDEA-(1–117)-(163–180) (red), showing the inclusion of aas 163–180 enhances LD localization and the ability to promote LD docking. The phenotypic distribution was performed in a minimum of three independent experiments for every construct (n>50 cells). HA signal in LDs was only detected in a proportion of the cells where HA-CIDEA constructs had induced LD enlargement or clustering, possibly due to the formation of CIDEA complexes reducing antibody accessibility to the HA epitope at the N-term. (**E**) A Hela cell expressing CIDEA-(F166R/V169R/L170R)-v5 (red) showing aa substitutions to compromise amphipathicity of the helix disrupt LD targeting, and a Hela cell expressing CIDEA-(K167E/R171E/R175/E)-v5 (red) showing amino acid (aa) substitutions to invert the charge of the helix but maintaining amphipathicity retains predominantly LD localization.

The following figure supplement is available for figure 3:

*Figure 3 continued on next page*

*Figure 3 continued*

**Figure supplement 1.** Conservation of amino acids (aas) for CIDEA amphipathic helix across vertebrate species.

In addition to the intrinsic differences between CIDEC and CIDEA, their activity could be affected by the interaction with additional proteins. While PLIN1 interacts with CIDEC, but not CIDEA, and potentiates its activity (*Sun et al., 2013*; *Grahn et al., 2013*), we have observed that CIDEA interacts with PLIN5 (*Figure 4D*), which is rich in BAT (*Harms and Seale, 2013*; *Zhou et al., 2003*). In addition to PLIN5, CIDEA showed high affinity for both CIDEB and CIDEC, while it did not co-IP with DFF40 or DFF45, which share homology with the N-term domain of CIDE proteins (*Figure 4C*). As BAT cells express high levels of both CIDEA and CIDEC, the formation of CIDE heterocomplexes could be involved in the regulation of LD enlargement to retain the multilocular state.

## CIDEA interacts with phosphatidic acid

To further characterize the interaction of CIDEA with the LD membrane, we utilized lipid strips to investigate the affinity of CIDEA for different lipids present in mammalian cell membranes and found that it selectively bound a set of anionic phospholipids (*Figure 5A*). The interaction with phosphatidic acid (PA) was of particular interest, as increased levels of this phospholipid have been linked with LD fusion (*Fei et al., 2011*) and the identification of enzymes such as AGPAT3 and LIPIN-1γ in LDs supports the existence of in situ generation and consumption of PA (*Wilfling et al., 2013*; *Wang et al., 2011*). PA binding was confirmed by the strong affinity of CIDEA-v5 for PA beads (*Figure 5B*), which was greatly reduced by pre-incubation of the lysate with soluble PA, but not phosphatidylcholine (PC). Although the N-term fragment showed some residual affinity, the main PA-binding site of CIDEA was in the C-term region containing the amphipathic helix (163–180) (*Figure 5C*). The charge inversion of its three cationic amino acids resulted in the loss of affinity for PA beads in the inactive mutant (K167E/R171E/R175E)-CIDEA-v5 without affecting its LD localization (*Figure 5C*), linking PA binding with the TAG-transference step (*Figure 5D*).

To investigate if PA affects the structure of the amphipathic helix, we repeated the CD analyses in phosphate buffer in the presence and absence of DLPC lipid vesicles with and without DLPA. Fitting of the CD data suggested a low (~5%) helical content for the wt peptide when analysed in phosphate buffer alone, and indicated a predominantly sheet/coil structure in the absence of detergent or liposomes. The presence of DLPC liposomes stabilized a sharp increase in α-helical structure (up to 40%) and an equivalent reduction of sheet content (*Figure 5E* and *Figure 5—figure supplement 1*), yielding higher helical content than that observed in n-dodecyl-β-D-maltopyranoside micelles (~25%, *Figure 3C*). In contrast, the induction of helix formation by DLPC was not observed in a mutant peptide carrying the substitutions impairing LD targeting in CIDEA-(F166R/V169R/L170R)-v5 (*Figure 5E*). This mutant peptide remained predominantly random coil in the absence and presence of DLPC liposomes. Interestingly, fitting of the CD data indicated significant helical content for both the wt and mutant peptides in the presence of DLPC:DLPA (9:1) vesicles (*Figure 5E* and *Figure 5—figure supplement 1*). This indicates that the interaction with the negatively charged phospholipid PA can compensate for the excess of positive charges in CIDEA-(F166R/V169R/L170R)-v5.

To obtain more detailed insight into the interaction of the amphipathic helix with phospholipids and the role of PA in this process, the interaction of the amphipathic helix with LDs was modelled using coarse-grained molecular dynamics (CG-MD) simulations (*Figure 5F–H*). CG-MD simulations are well established for lipid-containing systems (*Marrink et al., 2007*), including LDs (*Mohammadyani et al., 2014*), and have the advantage over full atomistic simulations in that the time scales required are much smaller allowing us to compare different LD compositions and helix mutants in the large multimolecular LD system. The wt (163–180) helix (CTSFKAVLRNLLRFMSYA) diffused towards the LD containing 400 palmitoyl-oleoyl-glycero-phosphocholine (POPC) and 200 TAG molecules where it interacted at its full length with the LD surface and penetrated into the hydrophobic region of the phospholipid monolayer covering the TAG core (*Figure 5F, G*). A similar behaviour was observed by the charge-inverted mutant K167E/R171E/R175E (*Figure 5G* and *Figure 5—figure supplement 2*), supporting the experimental result that these mutations do not affect LD localization in CIDEA (*Figures 2C* and *3E*). In contrast, no interaction with the LD was observed with the non-amphipathic F166R/L169R/V169R (*Figure 5G*), which also impairs LD binding in CIDEA

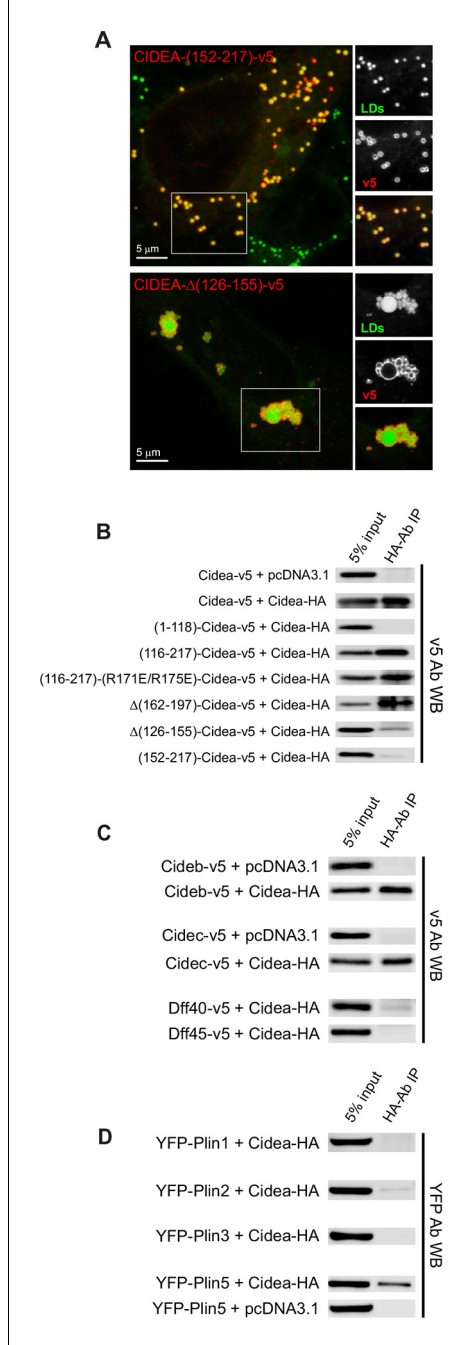

**Figure 4.** Liquid droplet (LD)–LD docking and CIDEA interactions. (**A**) A Hela cell expressing CIDEA-(152–217)-v5 showing normal recruitment to LDs, but no LD docking. A Hela cell expressing CIDEA-△(126–155)-v5 showing normal LD–LD docking but inefficient LD enlargement as revealed by the presence of clusters of small and large LDs. Representative images are shown of experiments performed in a minimum of three independent experiments for every construct (n>50 cells). (**B**) Co-immunoprecipitation (co-IP) assays between CIDEA-HA and different CIDEA-v5 constructs. The observed CIDEA–CIDEA interaction was driven by the C-term domain and required the presence of the 126–155 aa sequence. (**C, D**) Co-IP assays showing CIDEA interactions with CIDEB, CIDEC, and PLIN5. Each co-IP assay was performed at least in triplicate, producing similar results in each experiment.

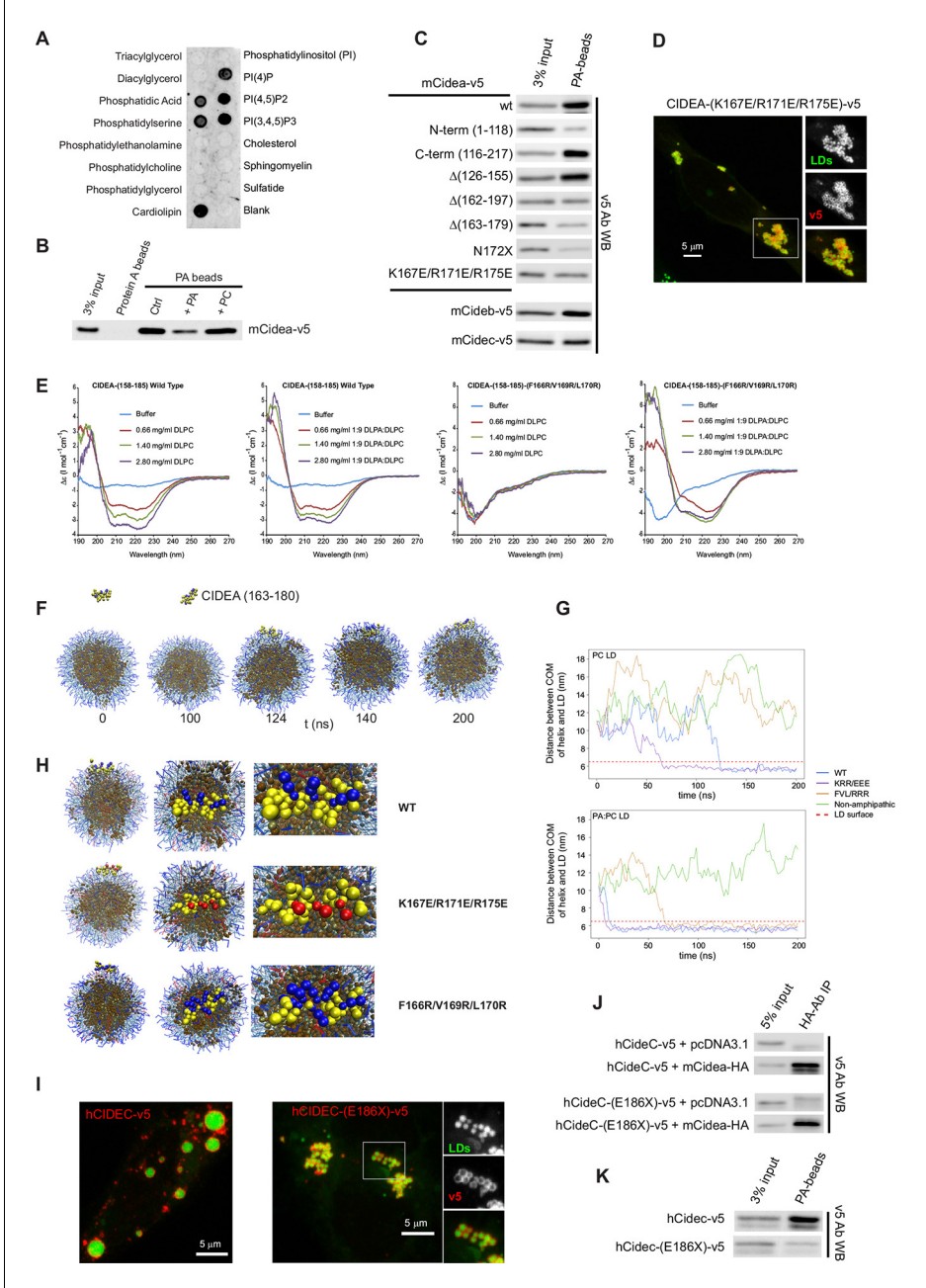

**Figure 5.** CIDEA is a phosphatidic acid (PA)-binding protein. (**A**) Lipid strip assay showing the affinity of CIDEA-v5 for certain anionic phospholipids. (**B**) Interaction of CIDEA-v5 with PA beads. Binding was reduced by pre-incubation of the lysate with soluble PA, but not phosphatidylcholine (PC). (**C**) The affinity for PA beads was highly reduced in CIDEA-v5 constructs with alterations in its C-term hydrophobic and basic region (162–197). (**D**) CIDEA-(K167E/R171E/R175E)-v5 localizes to LDs and induces their clustering but cannot promote their enlargement by lipid transfer. Representative images are shown of experiments performed in a minimum of three independent experiments for every construct (n>50 cells). (**E**) Circular dichroism spectra of the synthetic wild type (wt) or mutant (F166R/V169R/L170R) CIDEA peptides encompassing aas 158–185 solubilized in 25 mM sodium phosphate (pH 7.2) at concentrations of 70 µM (wt) and 47 µM (mutant). Peptide samples were prepared in the absence and presence of increasing amounts of DPLC or DLPC:DLPA (9:1 molar ratio). (**F–H**) Coarse-grained molecular dynamics (CG-MD) simulations of peptide interactions with LDs (PC: hydrophobic chains, transparent blue, polar heads, opaque blue; TAG: hydrophobic chains, dark brown, glycerol chain, light brown; PA: hydrophobic chains, transparent red, polar heads, opaque red; peptides: yellow, with cationic aa in blue and anionic in red). (**F**) Selected time points of the wt helix simulation with PC-LDs. At 124 ns the helix initiates the contact through its hydrophobic face, being rapidly embedded in the phospholipid monolayer. TAG molecules can abandon the neutral lipid core and are integrated in the hydrophobic region of the phospholipid monolayer. (**G**) Distance between the peptide and LD centre of mass (COM) versus time for the different helices with a PC-LD and a PC:PA-LD. The dashed line represents the approximate location of LD phospholipid head groups. (**H**) Different views of the configuration of the LD helix at the end of the simulations. Interaction

*Figure 5 continued on next page*

*Figure 5 continued*

between the polar head of PA and the helix can be observed for the wt and F166R/V169R/L170R but not K167E/R171E/R175E. (**I–K**) Comparison of full-length hCIDEC-v5 and the lipodystrophy-associated truncation hCIDEC-(E186X)-v5, including LD localization and morphology (**I**), co-IP with CIDEA-HA (**J**), and affinity for PA beads (**K**). Each co-IP, PA-binding assay, and lipid strip assay was performed at least in triplicate, producing similar results in each experiment.

The following figure supplements are available for figure 5:

**Figure supplement 1.** Secondary structure determination of CIDEA amino acids 158–185 (wild type and F166R/V169R/L170R) by CDPro DATABASE 4 (43 soluble proteins) using the CONTINLL program.

**Figure supplement 2.** Computational prediction of the amphipathic helix and LD interactions.

**Figure supplement 3.** TAG infiltration into the phospholipid monolayer.

**Figure supplement 4.** Computational prediction of PA docking to the amphipathic helix structure of CIDEA.

---

(*Figures 2C* and *3E*) and which was unable to attain stable secondary structure as evidenced by CD (*Figure 5E*). Similarly, a non-amphipathic α-helix in N-term (SSLQELISKTLDVLVITT) also showed no interaction with the LD (*Figure 5G*).

To study the effect of PA on LD structure and interaction with the CIDEA helix, we replaced 10% of the PC molecules with PA. The equilibration of the system resulted in a slight deformation of the spherical shape of the LD (*Figure 5—figure supplement 2*). The wt helix made a stable complex with this LD at even earlier simulation times than with the PC-only containing LDs (*Figure 5G* and *Figure 5—figure supplement 2*). The triple-E replacement mutant was also able to bind this LD, and even the F166R/L169R/V169R mutant was now able to interact with the membrane (*Figure 5G*). This result fits well with the CD results where the addition of PA also rescued the helix induction in this mutant peptide through interaction with the liposomes (*Figure 5E*). Interestingly, while the presence of PA permitted the accommodation of the F166R/L169R/V169R helix in the LD monolayer, it could not penetrate as deep towards the TAG core as the wt or helix. The average distance of the peptide to the centre of the LD ( ± SEM) was 5.6 ± 0.04 nm and 5.7 ± 0.03 nm for wt and K167E/R171E/R175E, respectively. In the presence of PA, the distance was 5.6 ± 0.02 nm, 5.7 ± 0.02 nm, and 6.1 ± 0.02 for wt, K167E/R171E/R175E, and F166R/L169R/V169R, respectively (also see *Figure 5G, H*). This result confirms that the presence of the hydrophobic face was necessary for proper helix insertion in the LD monolayer.

The CG-MD simulations not only shed light on the interaction between the helix and the LD, but also provided an indication of the mechanism by which this process could lead to LD enlargement by TAG transference. We observed that TAG molecules were able to escape the LD core and were integrated in the hydrophobic section of the membrane (*Figure 5H*). This TAG infiltration was increased after the docking of the wt helix in the membrane (*Figure 5—figure supplement 3*), suggesting that CIDEA could promote the migration of TAG into the membrane as an intermediate state prior to the transference to the acceptor LD. To complete the transference, the hydrophobic TAG molecules should overcome the energy barrier constituted by the phospholipid polar heads and water molecules in the LD–LD interface. Interestingly, we observed that the wt helix could attract PA molecules in its vicinity by the interaction of its cationic residues with the negatively charged polar head of PA (*Figure 5H*). A direct interaction of the amphipathic helix with PA was also indicated by molecular docking using Autodock Vina, which supported the role of the K167, R171, and R175 residues in the interaction (*Figure 5—figure supplement 4*). Remarkably, CG-MD simulations showed that whereas the non-amphipathic cationic helix F166R/L169R/V169R also interacted with PA molecules from its superficial docking position in the LD membrane, the anionic amphipathic mutant K167E/R171E/R175E was docked in a PA-depleted area and avoided the PA molecules (*Figure 5H*). Although TAG infiltration was also observed in this simulation and the helix was well embedded in the membrane, its inability to attract PA molecules could be responsible for the lack of TAG transference activity in CIDEA-(K167E/R171E/R175)-v5 (*Figure 5D*). Taken together, these results indicate that CIDEA binds the LD by embedding a cationic amphipathic helix into the

LD monolayer and that once there, it can interact with PA molecules, which could facilitate TAG transference.

We found that PA binding was a feature common to all three members of the CIDE protein family (*Figure 5C*). Intriguingly, we determined that an inactive CIDEC identified in a patient with lipodystrophy (*Rubio-Cabezas et al., 2009*) contained a truncation (E186X) in the predicted PA-binding site. Although hCIDEC-(E186X)-v5 and the equivalent mCIDEA-(N172X)-v5 were localized in LDs in a high percentage of cells, they were completely unable to induce LD enlargement (*Figures 2C* and *5I*). LD clustering activity and its ability to interact with CIDE proteins was not altered in hCIDEC-(E186X)-v5 (*Figure 5I, J*), but it showed no affinity for PA (*Figure 5K*). Thus, PA binding could be involved in the lipid transfer phase of CIDE activity.

## PA is required for LD enlargement

To confirm the requirement of PA binding, we examined the effect of PA depletion on CIDEA activity. While substantial alterations in the phospholipid composition of mammalian cells often compromise their viability, yeast cells offer a wide range of genetically modified strains with well-characterized alterations in phospholipid metabolism (*Figure 6A*) (*Henry et al., 2012*). Thus, despite the absence of CIDE homologues in yeast (*Wu et al., 2008*), we explored the functionality of CIDEA in wt and genetically modified strains of *Saccharomyces cerevisiae* (*Figure 6—source data 1*).

Murine CIDEA could be stably expressed in yeast cells (*Figure 6B*), producing an increase in the size of their LDs (*Figure 6C*). Yeast cells expressing wt CIDEA, but not the inactive R171E/R175E mutant, contained fewer and larger LDs than the control (*Figure 6D–F*), indicating that murine CIDEA was functional in these cells. By measuring the frequency of supersized LDs (diameter above 0.5 μm) and the total number of LDs in strains with altered lipid metabolism, we could determine the yeast strains in which CIDEA was able to induce LD enlargement (*Figure 6E, F*). CIDEA was inactive in cells defective in phospholipase D (pld1△) (*Rose et al., 1995*), which catalyzes the production of PA from PC. CIDEA activity was also abrogated in cells expressing a hyperactive form of the PA phosphohydrolase (PAH1-7A) (*Choi et al., 2010*; *Choi et al., 2012*). These results indicate that PA is necessary for CIDEA activity. In addition, we observed that total cellular PA levels were increased by CIDEA, an effect that was prevented by the expression of *PAH1-7A* (*Figure 6G*). As the PA synthesis rate was not affected (*Figure 6H*), CIDEA could be protecting a pool of PA from degradation.

The deletion of diacylglycerol kinase (dgk1△) (*Han et al., 2008*) did not affect CIDEA activity. *DGK1* is important for the generation of phospholipids from TAG as cells exit from stasis (*Fakas et al., 2010*), but its deletion has not been shown to have a great effect on PA levels under normal growth conditions. Regarding *PAH1*, its deletion produces dramatic cellular effects (*Santos-Rosa et al., 2005*), including defective LD formation (*Adeyo et al., 2011*; *Fakas et al., 2011*). As this alteration in LDs can be compensated by the deletion of *DGK1*, we chose to use the dgk1△pah1△ strain, observing normal LD enlargement by CIDEA (*Figure 6E, F*). CIDEA was also able to further increase the LD size in the cho2△ strain, which lacks the phosphatidylethanolamine (PE) methylation pathway for PC synthesis, and has been shown to present supersized LDs and high levels of PA and PE (*Fei et al., 2011*). The CIDEA-induced LD enlargement in yeast was not due to a mere coating effect protecting LDs against lipases, as it was functional in the tgl3△tgl4△tgl5△ strain, which lacks lipase activity. As expected, CIDEA could not induce the appearance of LDs in dga1△lro1△are1△are2△ cells (*Figure 6D*), which are deficient in the enzymes required for TAG and steryl ester synthesis and contain no LDs (*Sandager et al., 2002*).

To study the role of PA-dependent CIDEA action in mammalian cells without compromising other PA-dependent cellular processes, we specifically degraded this phospholipid in LDs by overexpressing a LD-localized isoform of PA phosphohydrolase (LIPIN-1γ) (*Wang et al., 2011*; *Han and Carman, 2010*). While CIDEA-HA displayed normal activity in cells co-transfected with an empty vector, it was unable to promote LD enlargement in cells expressing LIPIN-1γ-v5 (*Figure 6I, J*). LIPIN-1γ-v5 showed affinity for PA but it did not co-IP with CIDEA, indicating that its inhibitory effects were not due to a direct interaction between these proteins (*Figure 6K, L*). Taken together, these results reveal that the mechanism of action of CIDEA involves direct interaction with PA molecules in the LD monolayer.

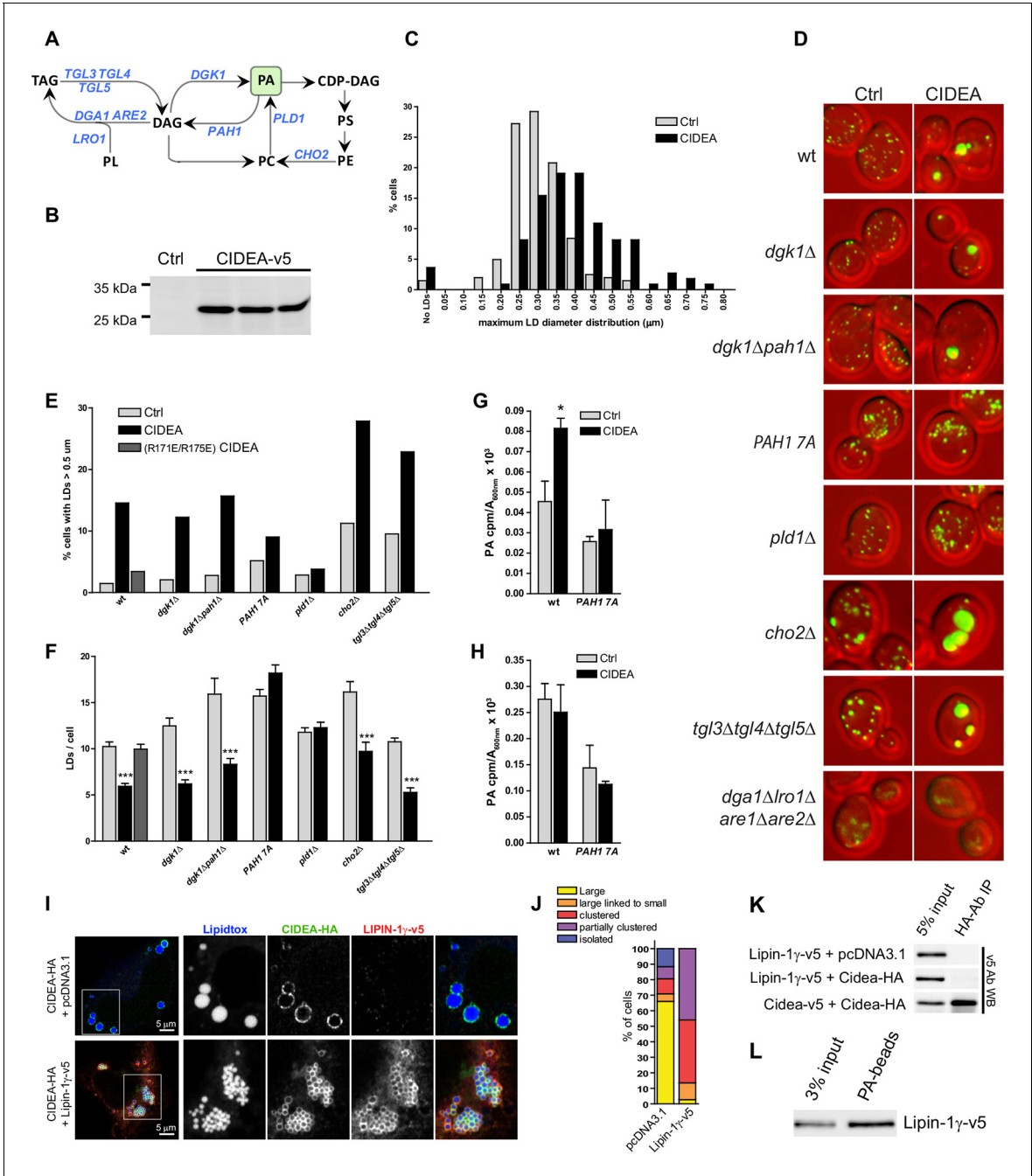

**Figure 6.** CIDEA is functional in yeast and requires PA. (A) Pathway showing the reactions catalyzed by the enzymes altered in the studied yeast strains. (B) Stable expression of mCIDEA-v5 in three transformed yeast clones. (C) Frequency distribution of the diameter of the largest LD per cell. (D) LD staining in the studied yeast strains transformed with pRS316-CYC1p-Cidea or the empty vector. (E, F) Quantification of LD size and number per cell in randomly acquired images (100–200 cells/condition). CIDEA activity in yeast was measured by its ability to increase the percentage of cells with supersized LDs (E) and reduce the total number of LDs per cell (F). (G, H) Effect of CIDEA and *PAH1*-7A expression in the cellular levels of PA (G) and its synthesis rate (H). Three different yeast clones per condition were analysed, and results are shown as the mean ± SEM. One-way ANOVA with Bonferroni post-test was performed to determine significant differences due to the presence of CIDEA (*$p<0.05$; ***$p<0.001$). (I–L) Coexpression of hLIPIN-1γ-v5 and CIDEA-HA in Hela cells. (I) Representative immunofluorescence images showing LD staining (blue) in Hela cells expressing CIDEA-HA (green) in the presence or absence of hLIPIN-1γ-v5 (red). Twenty-four hours after overexpression of hLIPIN-1γ-v5, cells were transfected with pcDNA3.1/Cidea-HA and incubated for a further 24 hr. (J) Phenotypic distribution in randomly selected cells (n>50) showing the average values for three independent experiments. (K) Co-IP assay in lysates of transfected Hela cells. (L) PA beads binding assay for hLIPIN-1γ-v5. Each co-IP and PA-binding assay was performed at least in triplicate, producing similar results in each experiment.

*Figure 6 continued on next page*

*Figure 6 continued*

The following source data is available for figure 6:

**Source data 1.** List of yeast strains used in this study.

## Discussion

The Cidea gene is highly expressed in BAT, induced in WAT following cold exposure (*Rosell et al., 2014*), and is widely used by researchers as a defining marker to discriminate brown or brite adipocytes from white adipocytes (*Harms and Seale, 2013*; *Zhou et al., 2003*). As evidence indicated a key role in the LD biology (*Hallberg et al., 2008*), we have characterized the mechanism by which CIDEA promotes LD enlargement, which involves the targeting of LDs, the docking of LD pairs, and the transference of lipids between them. The lipid transfer step requires the interaction of CIDEA and PA through a cationic amphipathic helix. Independently of PA binding, this helix is also responsible for anchoring CIDEA in the LD membrane. Finally, we demonstrate that the docking of LD pairs is driven by the formation of CIDEA complexes involving the N-term domain and a C-term interaction site.

CIDE proteins appeared during vertebrate evolution by the combination of an ancestor N-term domain and a LD-binding C-term domain (*Wu et al., 2008*). In spite of this, the full process of LD enlargement can be induced in yeast by the sole exogenous expression of CIDEA, indicating that in contrast to SNARE-triggered vesicle fusion, LD fusion by lipid transference does not require the coordination of multiple specific proteins (*Risselada and Grubmuller, 2012*). While vesicle fusion implies an intricate restructuring of the phospholipid bilayers, LD fusion is a spontaneous process that the cell has to prevent by tightly controlling their phospholipid composition (*Krahmer et al., 2011*). However, although phospholipid-modifying enzymes have been linked with the biogenesis of LDs (*Gubern et al., 2008*; *Andersson et al., 2006*), the implication of phospholipids in physiologic LD fusion processes has not been previously described.

Complete LD fusion by lipid transfer can last several hours, during which the participating LDs remain in contact. Our results indicate that both the N-term domain and a C-term dimerization site (aa 126–155) independently participate in the docking of LD pairs by forming trans interactions (*Figure 7*). Certain mutations in the dimerization sites that do not eliminate the interaction result in a decrease of the TAG transference efficiency, reflected by the presence of small LDs docked to enlarged LDs. This suggests that in addition to stabilizing the LD–LD interaction, the correct conformation of the CIDEA complexes is necessary for optimal TAG transfer. Furthermore, the formation of stable LD pairs is not sufficient to trigger LD fusion by lipid transfer. In fact, although LDs can be tightly packed in cultured adipocytes, no TAG transference across neighbour LDs is observed in the absence of CIDE proteins (*Gong et al., 2011*), showing that the phospholipid monolayer acts as a barrier impermeable to TAG. Our CG-MD simulations indicate that certain TAG molecules can escape the neutral lipid core of the LD and be integrated within the aliphatic chains of the phospholipid monolayer. This could be a transition state prior to the TAG transference, and our data indicate that the docking of the amphipathic helix in the LD membrane could facilitate this process. However, the infiltrated TAGs in LD membranes in the presence of mutant helices, or even in the absence of docking, suggests that this is not enough to complete the TAG transference.

To be transferred to the adjacent LD, the TAGs integrated in the hydrophobic region of the LD membrane should cross the energy barrier defined by the phospholipid polar heads, and the interaction of CIDEA with PA could play a role in this process, as suggested by the disruption of LD enlargement by the mutations preventing PA binding (K167E/R171E/R175E) and the inhibition of CIDEA after PA depletion. The minor effects observed with more conservative substitutions in the helix suggest that the presence of positive charges is sufficient to induce TAG transference by attracting anionic phospholipids present in the LD membrane. PA, whose requirement is indicated by our PA-depletion experiments, is a cone-shaped anionic phospholipid that could locally destabilize the LD monolayer by favouring a negative membrane curvature that is incompatible with the spherical LD morphology (*Kooijman et al., 2005*). Interestingly, while the zwitterion PC, the main component of the monolayer, stabilizes the LD structure (*Krahmer et al., 2011*), the negatively charged PA promotes their coalescence (*Fei et al., 2011*). This is supported by our CG-MD results

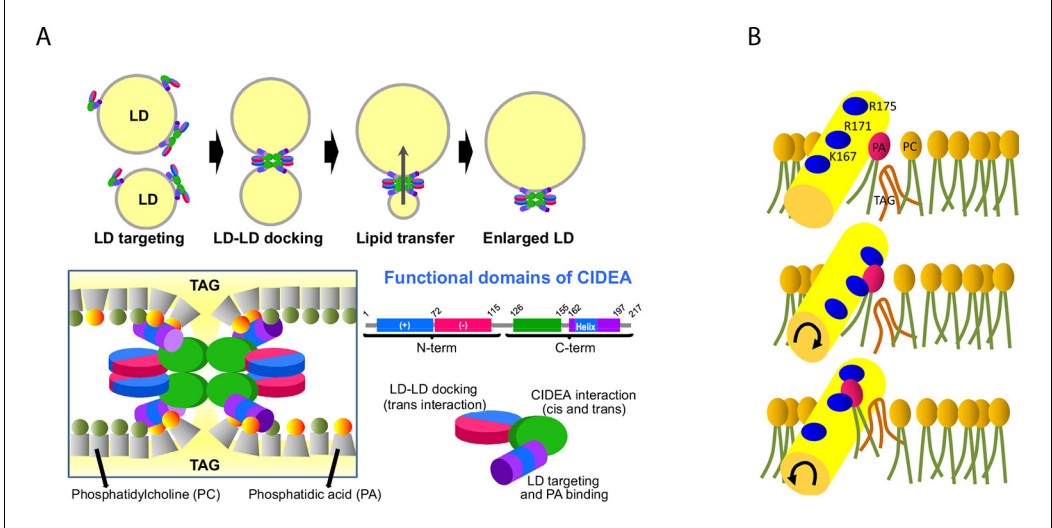

**Figure 7.** Proposed molecular mechanism. (**A**) CIDEA targets the LD through its C-term amphipathic helix and once diffused to the LD surface, it forms cis CIDEA complexes by interacting through the C-term (126–155) region. When two CIDEA-containing LDs make contact, trans interactions between CIDEA molecules in each droplet can be established, which will facilitate the docking of the LDs. Both the N-term and the C-term would contribute by dimerizing with their counterparts of the neighbour LD. This trans interaction will anchor the CIDEA complexes in the LD–LD contact site, promoting a local enrichment of CIDEA. The monolayers of the two LDs will be maintained at short distance by the CIDEA complex. The amphipathic helices, embedded in the hydrophobic region of the membrane, will interact with the cone-shaped PA, creating a local perturbation in the phospholipid barrier that will increase its permeability to TAG. (**B**) The docking of the amphipathic helix to the membrane could facilitate the integration of TAG molecules within the phospholipid hydrophobic tails. Although the helix will be stabilized with its cationic residues pointing outwards, it will interact with PA molecules in its vicinity, which could be pulled out of the monolayer by the helix molecular dynamics. This could create a transitory discontinuity in the polar barrier that will reduce the energy required to transfer the TAG molecules present in the membrane. This alteration, together with the microenvironment created by the CIDEA complex, will reduce the energy barrier necessary to transfer TAG molecules between LDs, allowing the LD growth by lipid transference.

which resulted in a deformation of the LD shape by the addition of PA. We propose a model in which the C-term amphipathic helix positions itself in the LD monolayer and interacts with PA molecules in its vicinity, which might include trans interactions with PA in the adjacent LD. The interaction with PA disturbs the integrity of the phospholipid barrier at the LD–LD interface, allowing the LD to LD transference of TAG molecules integrated in the LD membrane (*Figure 7*). Additional alterations in the LD composition could facilitate TAG transference, as differentiating adipocytes experience a reduction in saturated fatty acids in the LD phospholipids (*Arisawa et al., 2013*), and in their PC/PE ratio (*Hörl et al., 2011*), which could increase the permeability of the LD membranes; we previously observed that a change in the molecular structures of TAG results in an altered migration pattern to the LD surface (*Mohammadyani et al., 2014*).

During LD fusion by lipid transfer, the pressure gradient experienced by LDs favours TAG flux from small to large LDs (*Gong et al., 2011*). However, the implication of PA, a minor component of the LD membrane, could represent a control mechanism, as it is plausible that the cell could actively influence the TAG flux direction by differently regulating the levels of PA in large and small LDs, which could be controlled by the activity of enzymes such as AGPAT3 and LIPIN-1γ (*Wilfling et al., 2013*; *Wang et al., 2011*). This is a remarkable possibility, as a switch in the favoured TAG flux direction could promote the acquisition of a multilocular phenotype and facilitate the browning of WAT (*Barneda et al., 2013*). Interestingly, Cidea mRNA is the LD protein-encoding transcript that experiences the greatest increase during the cold-induced process by which multilocular BAT-like cells appear in WAT (*Barneda et al., 2013*). Furthermore, in BAT, cold exposure instigates a profound increase in CIDEA protein levels that is independent of transcriptional regulation (*Yu et al., 2015*). The profound increase in CIDEA is coincident with elevated lipolysis and de novo lipogenesis that occurs in both brown and white adipose tissues after β-adrenergic receptor activation (*Mottillo et al., 2014*). It is likely that CIDEA has a central role in coupling these processes to package newly synthesized TAG in LDs for subsequent lipolysis and fatty acid oxidation. Importantly,

BAT displays high levels of glycerol kinase activity (*Bertin, 1976*; *Bertin et al., 1984*) that facilitates glycerol recycling rather than release into the blood stream, following induction of lipolysis (*Portet et al., 1974*), which occurs in WAT. Hence, the reported elevated glycerol released from cells depleted of CIDEA (*Zhou et al., 2003*) is likely to be a result of decoupling lipolysis from the ability to efficiently store the products of lipogenesis in LDs, therefore producing a net increase in detected extracellular glycerol. This important role of CIDEA is supported by the marked depletion of TAG in the BAT of Cidea-null mice following overnight exposure to a temperature of 4°C (*Zhou et al., 2003*) and by our finding that CIDEA-dependent LD enlargement is maintained in a lipase-negative yeast strain.

Cidea and the genes that are required to facilitate high rates of lipolysis and lipogenesis are associated with the 'browning' of white fat either following cold exposure (*Rosell et al., 2014*) or in genetic models such as RIP140 knockout WAT (*Kiskinis et al., 2014*). The induction of a brown-like phenotype in WAT has potential benefits in the treatment and prevention of metabolic disorders (*Whittle et al., 2013*). Differences in the activity and regulation of CIDEC and CIDEA could also be responsible for the adoption of unilocular or multilocular phenotypes. In addition to their differential interaction with PLIN1 and 5, we have observed that CIDEC is more resilient to the deletion of the N-term than CIDEA, indicating that it may be less sensitive to regulatory post-translational modifications of this domain. This robustness of CIDEC activity together with its potentiation by PLIN1 could facilitate the continuity of the LD enlargement in white adipocytes until the unilocular phenotype is achieved. In contrast, in brown adipocytes expressing CIDEA the process would be stopped at the multilocular stage, for example, due to post-translational modifications that modulate the function or stability of the protein or alteration of the PA levels in LDs.

Further work will be required to characterize the physiological differences between CIDEC and CIDEA, and determine the influence of their interacting partners and the role of proteins that are able to alter the LD PA levels, such as Lipin-1γ. Abnormal accumulation of large LDs have also been observed in non-adipocyte cells under other pathological conditions such as liver steatosis and atherosclerosis (*Krahmer and Walther, 2013*). As enhanced expression of CIDE proteins have been linked to these conditions (*Li et al., 2010*; *Zhou et al., 2012*; *Matsusue et al., 2008*), the modulation of CIDE-triggered LD enlargement represents a potential therapeutic strategy that requires the elucidation of its molecular mechanism.

In summary, we found that during LD fusion by lipid transference, CIDEA ensures the close proximity of the LD membranes by forming trans complexes through its N-term and C-term dimerization sites. This protein complex will be anchored in the LD–LD interface, forming the molecular environment necessary for TAG transport across the membrane. Finally, the amphipathic helix embedded in the LD membrane interacts with the cone-shaped phospholipid PA, generating a local perturbation of the monolayer integrity that would increase its permeability to TAG and enable its exportation to the acceptor LD. The new mechanistic insight into the molecular events underpinning LD dynamics revealed by this study highlights CIDEA and PA production as targets for therapeutic modulation of LD accumulation.

## Materials and methods

### Plasmids and antibodies

The coding region of murine Cidea, Cideb, Cidec, Dff40, and Dff45 were cloned into the vector pcDNA3.1D/V5-His-TOPO (Invitrogen, Paisley, UK) to obtain the v5-tagged versions of the proteins (*Hallberg et al., 2008*). The human full-length and truncated forms of Cidec were subcloned into pcDNA3.1D/V5-His-TOPO from their GFP constructs (*Rubio-Cabezas et al., 2009*) and Lipin-1γ-v5 was constructed from pGH321 (*Han and Carman, 2010*). Mutations and deletions were generated with the QuikChange Lightning Kit (Agilent). Tagged proteins were detected by using antibodies against v5 (R96025; Invitrogen), HA (H6908; Sigma), or GFP (ab1218; Abcam).

### Cell culture and transfection

3T3-L1 cells were maintained in Dulbecco's modified Eagle's medium (DMEM) containing 4.5 g/l glucose and L-glutamine supplemented with 10% newborn calf serum (NCS; Invitrogen) and penicillin/streptomycin at 37°C and 5% $CO_2$. Hela cells were cultured in similar conditions but with 10% FBS

(Invitrogen). Transfections were performed using Lipofectamine 2000 (Invitrogen). Stable cell lines expressing CIDEA-v5 were generated by transfection of 3T3-L1 cells with pcDNA3.1/Cidea-v5, followed by selection with G418 (Invitrogen). The imBAT cell line was generated by the retroviral transduction of primary brown adipocytes with SV40 large-T antigen tsA58 mutant and differentiated as previously described (*Hallberg et al., 2008*).

## 4D live cell imaging

Cells in gelatin-coated glass bottom dishes were stained with 0.1–0.5 µg/ml BODIPY 493/503 in the appropriate culture medium with 20 mM HEPES in the absence of serum. After 10 min at 37°C, 10% FBS was restored and the dish was equilibrated at 37°C in a Leica SP5 confocal microscope. Time-lapse Z-stacks were acquired every 2 min and represented as their maximum projection. 3T3-L1 cells were analysed 6 hr after infection with an adenovirus vector expressing CIDEA (*Hallberg et al., 2008*). For the imBAT differentiation experiments, pre-adipocytes were incubated for 48 hr with differentiation cocktail (*Hallberg et al., 2008*), and medium was changed to DMEM:F12 with 10% FBS, 1nM T3, and 170 nM insulin for 6 hr before staining.

## Immunofluorescence

Cells on glass coverslips were fixed in 4% paraformaldehyde and permeabilized with blocking solution (BS: 0.5% BSA, 0.05% Saponin, 50 mM $NH_4Cl$ in PBS). Cells were incubated overnight at 4°C with primary antibodies diluted in BS, and for 1 hr at room temperature with secondary antibodies (conjugated to Alexa488 and Alexa555, Invitrogen). Cells were stained in PBS with 2 µg/ml BODIPY 493/503 or 1:200 dilution of LipidTox Deep Red for 15 min and mounted in ProLong Gold antifade reagent (all from Invitrogen). Images were acquired in a Leica TCS SP5 microscope. For the phenotypic distribution of Hela cells expressing modified CIDEA-v5 constructs, cells were treated with oleic acid 24 hr after transfection and incubated for a further 18 hr prior to fixation. Phenotype classification was performed by visual analysis of randomized samples in a minimum of three independent experiments for each construct (n>50 cells)

## Liposome preparation

Liposomes were prepared by dissolving lipid (1,2-dilauroyl-sn-glycero-3-phosphocholine (DLPC), 12:0 PC) or a mixture of DLPC and DLPA (1,2-dilauroyl-sn-glycero-3-phosphate, 12:0 PA; at a 9:1 DLPC:DLPA molar ratio) (Echelon Biosciences, USA) in 3:1 chloroform:MeOH and drying under vacuum using rotary evaporation. The resulting thin films were left to dry under vacuum overnight to remove all residual solvent, reconstituted in 25 mM sodium phosphate buffer (pH 7.2) to a final lipid concentration of 3.3 mg/mL, and subjected to four times freeze-thaw-sonicate cycles. The vesicles were incubated at 37°C for 20 min prior to CD measurements.

## Circular dichroism

CD experiments were undertaken with a synthetic wt (SYDIRCTSFKAVLRNLLRFMSYAAQMTG) CIDEA peptide (Pepmic, Suzhou, China) encompassing aas 158–185 solubilized at a concentration of 41 µM (based on absorbance at 280 nm) in 50 mM potassium phosphate, pH 6.2 plus 0.1% n-dodecyl-β-D-maltopyranoside and analysed by CD in a Jasco J-815 spectrometer (Jasco UK, Great Dunmow, UK).

Additional CD experiments with the same wt peptide and a mutant (F166R/V169R/L170R) (SYDIR-CTSRKARRRNLLRFMSYAAQMTG) were carried out using a Jasco J-1500 spectropolarimeter (Jasco UK) equipped with a Peltier thermally controlled cuvette holder and 1 mm path-length quartz cuvettes (Starna; Optiglass, Hainault, UK). Spectra were recorded between 190 and 300 nm with a data pitch of 0.2 nm, a bandwidth of 2 nm, a scanning speed of 100 nm min$^{-1}$ and a response time of 1 second. Peptides were solubilized in 25 mM sodium phosphate (pH 7.2) at concentrations of 70 µM (wt) and 47 µM (mutant). Peptide samples were prepared in the absence and presence of DPLC and DLPC:DLPA (9:1 molar ratio) vesicles and CD spectra were acquired at 37°C. Data shown were averaged from four individual spectra after subtraction of the appropriate buffer/vesicle CD spectrum. All CD data were analysed using the CDPro suite of programs. The output of the individual programs CDSSTR and CONTINLL provided the estimated percentages of α-helix, β-sheet, turn, and

unstructured regions, using the IB = 4 database of 43 soluble proteins with CD data from 190–250 nm.

## Structure prediction and molecular docking

Secondary structure propensity of full-length CIDEA was predicted using DSSP (*Arnold et al., 2006*). The amphipathic helix sequence CTSFKAVLRNLLRFMSYA (163–180 aa) was submitted to the PEP-FOLD online de novo peptide structure prediction server using default settings (*Maupetit et al., 2009*). PA was docked to the PEP-FOLD predicted structure using default settings in a single simulation by AutoDock Vina53 (http://vina.scripps.edu) (*Trott et al., 2009*). Lipid and protein structures were converted from pdb into pdbqt format using MGL Tools54. A grid box was centred at coordinates 35.651, 35.471, 35.569 with 34 Å units in x, y, and z directions to cover the entire helix. AutoDock Vina reports the nine lowest energy conformations, which were inspected using PyMOL software (www.pymol.org). According to binding affinity and visual inspection, without RMSD clustering, the best-fit model has been selected.

## CG-MD simulations

CG-MD simulations were used to predict the structure of a LD and its putative interaction with the amphipathic helix using a 4 to 1 atom mapping for both, lipids and protein (*Marrink et al., 2007*; *Monticelli et al., 2008*). A LD composed of a hydrophobic core containing 200 glyceryl trioleate or TAG molecules surrounded by a phospholipid monolayer containing 400 POPC molecules previously reported was used as the starting configuration (*Mohammadyani et al., 2014*). A second LD containing PA consisting of a hydrophobic core of 200 TAG molecules, and a phospholipid monolayer with 364 POPC molecules and 36 palmitoyl-oleoyl-glycero-phosphatidic acid (POPA) was compiled using the same procedure. A rectangular simulation box including LD, amphipathic helix, water, and ions was energy minimized and pre-equilibrated. All MD runs were carried out for 200 ns under NPT conditions. The CG-MD simulation of the LD–helix interaction was carried out using the MARTINI CG force field developed by Marrink et al. (version 2.0) (*Marrink et al., 2007*). All simulations were performed using the GROMACS simulation package version 4.6.5 (http://www.gromacs.org/). The system was weakly coupled to an external temperature bath at 310 K (*Berendsen et al., 1984*). The pressure was weakly coupled to an external bath at 1 bar using an isotropic pressure scheme (*Berendsen et al., 1984*). Visualization and analysis was performed using the VMD v.1.9 visualization software (*Humphrey et al., 1996*). Distances and density maps were computed using analysis tools (g_dist and g_densmap) in the GROMACS package (http://www.gromacs.org) (*Van Der Spoel et al., 2005*).

## Immunoprecipitation

Cells were lysed in 50 mM Tris (pH 8.0), 150 mM NaCl, 1% TRITON X-100 with protease inhibitor cocktail (Roche). Anti-HA antibody (H6908; Sigma) or anti-V5 antibody (R96025; Invitrogen) was bound to Dynabeads Protein G (Invitrogen) and incubated with the lysate to immunoprecipitate the tagged proteins following manufacturer's instructions. Cell lysates or IP fractions in Laemmli buffer were analysed by Western blot. Each co-IP experiment was performed at least in triplicate, producing similar results in each experiment with a representative image presented.

## Lipid binding assays

In vitro translated CIDEA-v5 was synthesized from pcDNA3.1/Cidea-v5 using the TnT Coupled Wheat Germ Extract System (Promega) and verified by Western blot. The cell-free preparation of CIDEA-v5 was probed with Membrane Lipid Strips (Echelon Biosciences) following the manufacturer's instructions. Protein affinity for PA was examined in pull-down assays using PA covalently linked to agarose beads (PA beads) (*Manifava et al., 2001*). Cells were lysed in 50 mM Tris-HCl pH 8.0, 50 mM KCl, 10 mM EDTA, 0.5% Nonidet P-40, and protease inhibitors. Lysates were sonicated and centrifuged at 14000*g* prior to incubation with the PA beads as previously described (*Manifava et al., 2001*). Competition experiments with soluble phospholipids were performed by supplementing the cell lysate with 1,2-dilauroyl-sn-glycero-3-phosphate 12:0 PC (DLPA) or 1,2-dilauroyl-sn-glycero-3-phosphocholine (DLPC) (Echelon Biosciences). Each PA-binding experiment

was performed at least in triplicate, producing similar results in each experiment with a representative image presented.

## CIDEA expression in yeast

The *S. cerevisiae* strains used in this study are listed in Supplementary file 1. To express CIDEA in yeast, a codon-optimized version of the mouse Cidea gene was generated by artificial gene synthesis (GeneOracle), and subcloned into pRS316-CYC1p. Wt BY4742 (*Brachmann et al., 1998*) and genetically modified yeast strains were transformed with pRS316-CYC1p-Cidea and stable transformants were selected in synthetic media minus uracil. Leucine selection was used for the expression of PAH1-7A with pHC204 (*Choi et al., 2010*).

## Microscopy analysis of yeast lipid droplets and image processing

Yeast cells in synthetic media cultured overnight at 30°C were diluted to OD600 = 0.1 and allowed to grow until mid-logarithmic phase (OD600 = 0.5) before fixation with 4% formaldehyde and LD staining with 2 µg/ml BODIPY 493/503. For the automatic quantification of LDs, random microscopy images were acquired using a Delta Vision RT system (Applied Precision). Maximum intensity and integrated intensity projections were created from the deconvolved image stacks using ImageJ. A custom written CellProfiler pipeline (*Carpenter et al., 2006*) automatically identified individual yeast cells and measured their number and size of circle shaped LDs. Supersized LDs were defined as the LDs with a diameter above 0.5 µm.

## Steady state and pulse labelling of phospholipids

To measure the total levels of phospholipids in yeast, cells were grown overnight in synthetic medium at 30°C in the presence of 20 µCi/mL [$^{32}$P]-orthophosphate. Cultures were then diluted to OD600 = 0.1 maintaining the label and were allowed to grow until OD600 = 0.5. To analyse *de novo* synthesis of glycerophospholipids, cells were grown to OD600 = 0.5 in synthetic medium and incubated with 100 µCi/mL [$^{32}$P]-for 20 min. Lipids were extracted and quantified by two-dimensional chromatography, as described by (*Gaspar et al., 2006*).

## Acknowledgements

We thank Dr Carole Sztalryd for providing the Plin expression vectors, Dr David Savage for the hCIDEC constructs, and Prof. Parmjit Jat for providing retrovirus to express the temperature-sensitive SV40 large T antigen. We are also grateful to Dr Vishwajeet Puri for critical reading of the manuscript. This work was supported by the BBSRC grant BB/H020233/1, the EU FP7 project DIABAT (HEALTH-F2-2011-278373), the Genesis Research Trust and by National Institutes of Health grants GM-19629 (to SAH) and GM028140 (to GMC).

## Additional information

### Funding

| Funder | Grant reference number | Author |
| --- | --- | --- |
| Biotechnology and Biological Sciences Research Council | BB/H020233/1 | David Barneda Mark Christian |
| European Commission | DIABAT (HEALTH-F2-2011-278373) | Mark Christian |
| National Institutes of Health | GM- 19629 | Susan A. Henry |
| Genesis Research Trust | | Mark Christian |
| National Institutes of Health | GM028140 | George M Carman |

The funders had no role in study design, data collection and interpretation, or the decision to submit the work for publication.

## Author contributions

DB, MLG, JKS, SAH, Conception and design, Acquisition of data, Analysis and interpretation of data, Drafting or revising the article; JPI, Computer Modeling, Conception and design, Acquisition of data, Analysis and interpretation of data, Drafting or revising the article; DM, DD, G-SH, SAJ, GMC, Conception and design, Acquisition of data, Analysis and interpretation of data; SP, Performed Circular Dichroism spectrometry, Conception and design, Acquisition of data, Analysis and interpretation of data; VK, MGP, Conception and design, Analysis and interpretation of data, Drafting or revising the article; NTK, Conception and design, Analysis and interpretation of data, Contributed unpublished essential data or reagents; AMD, CD Spectrometry, Conception and design, Acquisition of data, Analysis and interpretation of data, Drafting or revising the article; MC, Conception and design Acquisition of data Analysis and interpretation of data Drafting or revising the article

## Author ORCIDs

Dariush Mohammadyani, http://orcid.org/0000-0002-0505-779X

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
