## [Decision Letter]

Thank you for sending your work entitled "The brown adipocyte protein CIDEA promotes lipid droplet fusion via a phosphatidic acid-binding amphipathic helix" for consideration at *eLife*. Your article has been favorably evaluated by Vivek Malhotra (Senior editor) and three reviewers, one of whom is a member of our Board of Reviewing Editors. All the reviewers were experts in lipid-binding proteins, and two were experts in the structural biology of protein-lipid interactions.

All three reviewers thought that this manuscript was interesting and that it contained novel and important information governing the biology of lipid droplets in adipocytes. The authors propose that the growth of lipid droplets involves the binding of an amphipathic helix in CIDEA to phosphatidic acid (PA), and that LD pairs are docked by CIDEA trans complexes through contributions of the N-terminal domain and a C-terminal dimerization region. All three reviewers had substantial concerns about the function of the amphipathic helix in CIDEA. Their concerns are listed below and should be addressed. Specifically:

1) The authors should provide more information regarding the function of this domain, particularly the specificity of the interaction with phosphatidic acid. Many criticisms center on this issue.

2) The reviewers were uncertain about the validity of the experimental systems that were used in this paper. All three reviewers wanted to see experiments that would directly assess the binding of the amphipathic helix (as well as mutant amphipathic helices) to a lipid monolayer at an aqueous interface. The authors should either use liposomes of defined composition or use a variety of oil-water interface systems that have been used to study the biophysical properties of amphipathic helices in other proteins, for example plasma apolipoproteins. These additional studies are important.

3) The manuscript has a very large amount of information, reflecting considerable work. Many of the figures are impressive. However, it would be helpful to specify how many times each experiment was repeated.

4) Also, with respect to the western blots, most of them showed only "cut-out" bands, and it was not clear whether those bands were taken from the same experiment and the same western blot. Please clarify. And if the bands were taken from different experiments, please discuss how the studies were interpreted.

5) Also, as noted by the reviewers, several of the experiments contained an insufficient number of controls.

Reviewer 1:

CIDEA is a lipid droplet (LD)-protein enriched in brown adipocytes promoting the enlargement of LDs which are dynamic, ubiquitous organelles specialized for storing neutral lipids. The authors propose that this process involves phosphatidic acid (PA) binding to an amphipathic helix in CIDEA, and that LD pairs are docked by CIDEA trans complexes through contributions of the N-terminal domain and a C-terminal dimerization region.

Overall, this is an excellent to outstanding paper, which reveals new mechanisms in the formation of lipid droplets in adipocytes. The notion that the CIDEA amphipathic helix interacts avidly and specifically to phosphatidic acid is critical for this paper. I believe that the authors should make an attempt to document that the binding of CIDEA to a phospholipid-water interface (with or without triolein) interface and document changes in surface pressure, with the goal of documenting the specificity of the interaction with PA.

Some of the experiments need additional controls. For example, I would like the authors to better document the CD spectra of the mutant helix containing the Glu mutations. How about showing data for the mutant helix in Figure 2?

Also, this reviewer would like to know more information about the specificity of the amphipathic helix interactions with PA. Would the interaction occur if the Arg residues were replaced with Lys or His, and if the Lys residue were replaced with Arg or His? Please report the conservation of all residues within the helix in vertebrate evolution.

Much of the data in the paper appears to be bands cut out of a western. Were all of these experiments performed on the same gel, or multiple gels? If the latter, please explain the controls for each experiment.

In Figure 4, it would be nice to show additional controls with other anionic phospholipids bound to beads – and examine competition with other anionic phospholipids. Also, testing the CIDEA mutant with the Glu mutations would be helpful. Similarly, Figure 4 and Figure 5 need additional controls.

Reviewer 2:

An important question that had remained open in research on CIDEA concerned the mechanistic on how CIDEA is involved in a hypothesized deformation of the monolayer of phospholipids, which surround the neutral lipid core of LD, and therefore might facilitate the transfer of neutral lipids.

Barneda et al. could identify mutants in the N-terminal half of the N-terminal domain to result in high percentages of large clustered LD with inefficient TAG transfer. The CIDE-N domain also appears to be involved in LD-LD docking since entire deletion of this domain or large parts of it yield mainly small, isolated lipid droplets). Additionally they identified an amphiphatic helix in the C-terminal half which leads to small, yet highly clustered LD indicating a defect in transfer of neutral lipids. The authors attribute this' LD phenotype' to impaired PA binding.

General comments:

Figure 1–figure supplement 1 is a central figure and should be included in the main text.

Introduction and subsection “A cationic amphipathic helix in C-term drives LD targeting”: I would appreciate a more conservative approach in separating previously existing knowledge from new findings presented in this manuscript (e.g. of 3D structures or LD binding regions).

Subheading “A cationic amphipathic helix in C-term drives LD targeting” and Figure 2: According to the hypothesis put forward by the authors, the short sequence 162-197 should target the N-terminal domain fused to the LD targeting helix to the LD. This should be shown experimentally and would strongly enhance the sole importance of this helix as targeting signal.

Subheading “A cationic amphipathic helix in C-term drives LD targeting” and Figure 2: CD completely lacks unit in x- and y-axis. Assuming that y-axis corresponds to MRE the negative peaks at 209nm appear to be extremely strong and beyond -35000. Are concentrations correct? The computed numbers of the CD spectrum should be related to the length of the measures peptide. What does 4% b-sheet mean in context of an 18-residue peptide?

Paragraph two, subheading “A cationic amphipathic helix in C-term drives LD targeting” and Figure 2: Is the identified stretch of 18 amino acids embedded in a larger helix? Figure 2 gives the impression of a positive charge and negative charge at the N-and C-term of the helix respectively. Did the authors take care of the fact, that these are not charged NH3+ and COO- groups in context of the entire protein in their figures and MD calculations?

Paragraph two, subheading “A cationic amphipathic helix in C-term drives LD targeting”: The in-silico studies show a representation of the process that the helix diffuses into the micelles (assuming that the helix is already formed in the aqueous phase). However, since models were used I would be extremely careful making quantitative statements as "high affinity" or "quickly diffused" since they are not associated with quantitative numbers. These conclusions would require much more calculations as presented in this paper. Does an arbitrary 'control' helix e.g. from the N-terminal domain which is not involved in LD targeting not diffuse into the micelle or at a significantly lower pace?

Subheading “LD-LD docking is induced by the formation of CIDEA complexes”: The entire section puts forward very strong conclusions based on just one type of experiments. I suggest that other techniques could be used to test the protein-protein interaction.

Paragraph three of the aforementioned subsection: I do not see the CoIP of CIDEA in Figure 3.

Single point mutants in the N-terminal domain (R47E, E79Q/D80N) also show major phenotypic effects and should be discussed in more detail, especially in light that 3D structures with high sequence identify in these region are available. Interestingly, the LD pattern of the variants R47E (Figure 3) compared to R47A and R47Q (Figure 1–figure supplement 1) is drastically different and should be discussed in more detail.

Reviewer 3:

The models developed in this manuscript could explain a lot about brown versus white fat. The concept of physical-chemical based models to explain switching between small FD in BAT and large FD in WAT is reasonably compelling but the work needs more convincing work on supporting the existence of a functioning amphipathic helix and its proposed interactions with PA on the surface of FD or TAG emulsions.

Suggested additional experiments:

1) Prove interactions of AH, native sequence and lack of interactions of mutated control sequences, with TAG emulsions. The CD data of peptide in solvent is of no help here. If an experiment showing increased helicity with lipid binding can be devised this will be helpful also.

2) Show that interactions with TAG emulsions increase (or otherwise modified) with addition of PA to surface monolayers.

---

## [Author Response]

All three reviewers thought that this manuscript was interesting and that it contained novel and important information governing the biology of lipid droplets in adipocytes. The authors propose that the growth of lipid droplets involves the binding of an amphipathic helix in CIDEA to phosphatidic acid (PA), and that LD pairs are docked by CIDEA trans complexes through contributions of the N-terminal domain and a C-terminal dimerization region. All three reviewers had substantial concerns about the function of the amphipathic helix in CIDEA. Their concerns are listed below and should be addressed. Specifically:

*1) The authors should provide more information regarding the function of this domain, particularly the specificity of the interaction with phosphatidic acid. Many criticisms center on this issue.*

The identification and functional characterization of the amphipathic helix in CIDEA C-term is one of the major strengths of our manuscript and in this revised form we provide substantial new data to further clarify its role in LD targeting and LD enlargement.

The role of the amphipathic helix controlling the LD localization of CIDEA has been further explored using complementary approaches following the reviewers’ suggestions: i) By CD spectroscopy we have determined the stabilization of the helical structure through interaction with liposomes of different composition. ii) We have extended our CG-MD simulations by analysing a range of mutant peptides with LDs containing or lacking PA, showing a remarkable concordance with our in vitro and cellular data. iii) We have demonstrated that the conjugation with the 18-amino acid sequence is sufficient to target the soluble N-term fragment of CIDEA into LDs and induce their clustering.

Whereas specific protein-lipid interactions can control the selective membrane translocation of soluble proteins to certain organelles, we do not claim that the interaction of CIDEA and the LD membrane is mediated by PA-binding. In fact, our data clearly indicate that LD-targeting is independent of PA binding. Firstly, we found that some mutants that are incapable of binding PA and promoting LD enlargement can efficiently target the LDs. Secondly, the depletion of PA in LDs by LIPIN-1γ prevents CIDEA-induced LD enlargement but does not affect the LD localization of CIDEA. Thirdly, our CD analysis and GC-MD simulations reveal that interaction with liposomes or modelled LDs does not require the presence of PA.

In addition to its role in LD-targeting, we demonstrate that the amphipathic helix is essential for the LD enlargement step and that this process depends on the interaction of CIDEA with PA through this domain. It is important to emphasise that we are not claiming that CIDEA only binds PA as in this paper we show that it can interact with other phospholipids such as cardiolipin or phosphatidylinositol phosphates. We have focused on PA due to its suggested role in modulating membrane dynamics to facilitate LD enlargement and the recent identification in LDs of enzymes that generate and consume PA (AGPAT3 and LIPIN-1γ). Importantly, our data unequivocally demonstrate that CIDEA can interact with PA and that the presence of PA is necessary for CIDEA activity. PA depletion in yeast and mammalian cell models prevents LD enlargement without affecting the LD localization of CIDEA, and we demonstrate that the amphipathic helix is essential for PA-binding and LD enlargement, as its disruption by truncation of the protein or amino acid substitutions results in the prevention of both processes. Finally, the additional CG-MD simulations of the PA-helix interaction in the context of the LD membrane have been utilized to clarify the potential role of PA in the molecular mechanism of CIDEA action (detailed in Figure 7).

*2) The reviewers were uncertain about the validity of the experimental systems that were used in this paper. All three reviewers wanted to see experiments that would directly assess the binding of the amphipathic helix (as well as mutant amphipathic helices) to a lipid monolayer at an aqueous interface. The authors should either use liposomes of defined composition or use a variety of oil-water interface systems that have been used to study the biophysical properties of amphipathic helices in other proteins, for example plasma apolipoproteins. These additional studies are important.*

We have instigated successful new biophysical investigations of interactions between CIDEA and lipids. For this we have undertaken analysis of CD spectra of the CIDEA amphipathic helix and a non-amphipathic mutant in aqueous solution and then in the presence of liposomes of different compositions.

In addition we have substantially increased our range of CG-MD simulations, modelling the interaction of wild type and mutant helices with LDs composed of PC:TAG (400:200) and PC:PA:TAG (364:36:200). The different dynamics observed remarkably fit the results of our CD analysis of the peptide interaction with liposomes and the cellular phenotype induced by the different CIDEA mutants, confirming the validity of these simulations for the study of CIDE activity. This has offered novel insights on the nature of CIDEA interaction with PA, complementing our model for the mechanism of action of CIDEA.

*3) The manuscript has a very large amount of information, reflecting considerable work. Many of the figures are impressive. However, it would be helpful to specify how many times each experiment was repeated.*

We have amended the legends and Methods section in the revised manuscript to clarify the number of times and replicates that were performed for each experiment. We have also clearly stated the number of transfected cells assessed for lipid droplet morphology by confocal microscopy per plasmid. The phenotypic distribution was performed in a minimum of three independent experiments for every construct (n > 50 cells). Although the selection of microscopy images representative of the main phenotype of transfected cells is a common and valid practice in cell biology, we believe our approach, based on the quantification of the different phenotypes observed, has proved more reliable and informative. Together with the analysis of a large number of mutants this has permitted the identification and fine mapping of the different phases of CIDEA activity, providing novel insights for each of these processes:

LD targeting: we demonstrate that CIDEA is docked to the LD membranes by the insertion of an amphipathic helix.

LD-LD docking: by optimising our cellular model to minimize LD clustering independent of CIDEA-CIDEA interactions we have been able to functionally demonstrate the role of the N-term and C-term dimerization sites previously identified in CIDE proteins.

LD enlargement: we demonstrate that independently of its role in LD targeting, the amphipathic helix controls LD enlargement in a process that requires its interaction with PA.

*4) Also, with respect to the western blots, most of them showed only "cut-out" bands, and it was not clear whether those bands were taken from the same experiment and the same western blot. Please clarify. And if the bands were taken from different experiments, please discuss how the studies were interpreted.*

The Western Blots referred to by the reviewers addressed protein interactions of wild type and proteins with truncations or amino acid substitutions. Separate Western blots were performed for the different expressed proteins. Our Western Blot analysis is completely valid for assessing protein interactions as each individual protein was always analysed on the same blot as the input of the same protein. The input protein confirms that the exogenously expressed protein was both successfully expressed and detected by Western Blotting and thus, we can definitively determine if there was a direct protein-protein interaction or not. Importantly, for the Co-IPs, where a positive interaction is shown, we have included negative controls, namely IPs from transfected cells in which the epitope-tagged protein targeted by the antibody has been replaced by an empty vector. As no protein was IPed with these controls, it shows that the protein not targeted by the antibody was not capable of being pulled down in the absence of its interacting partner. Therefore, the Co-IPed proteins are only pulled down and detected by Western Blot when a true interaction is present.

*5) Also, as noted by the reviewers, several of the experiments contained an insufficient number of controls.*

In addition to developing a new assay for CIDEA CD spectrometry we have included the additional control of the peptide with substitutions F166R/V169R/L170R which was designed to disrupt its amphipathic nature. Our new data showed that the WT CIDEA amphipathic helix folded in the presence of DLPC liposomes whereas the peptide with amino acid substitutions under the same conditions was not capable of folding to an alpha-helix.

More controls were also requested to demonstrate the specificity of the protein interaction with PA. However, we have clearly shown a physical interaction between CIDEA and PA as well as a functional requirement for the interaction. Furthermore, additional investigations of binding specificity are not a requirement of our mechanistic model and we actually show that CIDEA can interact with other anionic phospholipids. As claimed in the text:

“We utilized lipid strips to investigate the affinity of CIDEA for different lipids present in mammalian cell membranes, finding that it selectively bound a set of anionic phospholipids (Figure 5)”.

Due to its relevance in LDs, the interaction with PA was further analysed using agarose-PA beads, a method that was validated by comparison with agarose-Protein A beads and by competition with soluble PC and PA. Once demonstrated its reliability, the assay was extended to additional CIDE constructs. The functional requirement of PA for CIDEA action was demonstrated in yeast and mammalian cells (Figure 5 and Figure 6).

For the CG-MD simulations, we agree that additional controls were necessary. In addition to studying the effect of PA and eliminating the peptide terminal charges, we have included simulations with mutant and N-term helices to measure their behaviour in relation to that observed for the wild type.

Reviewer 1:

*[…] Overall, this is an excellent to outstanding paper, which reveals new mechanisms in the formation of lipid droplets in adipocytes. The notion that the CIDEA amphipathic helix interacts avidly and specifically to phosphatidic acid is critical for this paper. I believe that the authors should make an attempt to document that the binding of CIDEA to a phospholipid-water interface (with or without triolein) interface and document changes in surface pressure, with the goal of documenting the specificity of the interaction with PA.*

We would like to clarify that we have identified that the LD-enlarging function of CIDEA is dependent on the binding of PA to an amphipathic helix. However, we show that CIDEA can interact with a range of other phospholipids and our proposed mechanism does not exclude that other negatively charged lipids present in LD membranes could contribute to CIDEA activity. However, the key role of PA is clearly demonstrated in the PA depletion experiments, which reveal the functional dependence of CIDEA on PA levels. Further details in our response to the role PA interaction in CIDEA function are outlined in our response to the editor’s comments (comment 1).

The measurement of changes in surface pressure in a phospholipid-water interface will help the characterization of the LD-binding process but is beyond the scope of this paper. However, with our new CD assays in the presence or absence of liposomes, we offer alternative biophysical evidence of the interaction between the amphipathic helix and phospholipid membranes. As expected, in line with our previous results showing that LD-targeting and PA-binding are independent processes, the presence of PA in the liposomes was not required for the stabilization of the peptide’s helical structure.

Some of the experiments need additional controls. For example, I would like the authors to better document the CD spectra of the mutant helix containing the Glu mutations. How about showing data for the mutant helix in Figure 2?

We considered that the Glu mutations (K167E/R171E/R175E) would not offer an ideal control for the CD experiments. Whereas these mutations prevent LD enlargement and PA-binding, they do not influence the ability of CIDEA to target the LD membranes. In addition, as we did not observe significant differences in the CD profile of the wt helix by the addition of a 10% PA to the PC liposomes, we considered that the analysis of this PA-binding defective mutant would not offer additional information, other to confirm its unaffected folding and interaction with membranes. In contrast, as the CD assay proved a useful tool to analyse the interaction of the amphipathic helix with membranes (regardless of the presence of PA), we decided to analyse the effect of amino acid substitutions impairing LD targeting. In view of the previous results with CIDEA-(L169R/V170R)-v5, we added CIDEA-(F166R/L169R/V170R)-v5 to the LD phenotype analysis, observing impaired LD binding. These amino acid substitutions eliminate the amphipathic nature of the helix, by introducing positive charges in its hydrophobic face. Therefore, we chose the (F166R/V169R/L170R) peptide as a control for our new CD analysis. The new data represent a significant addition to the previous analysis by demonstrating that in the presence of DLPC liposomes, the wild type peptide shows enhanced alpha helical folding whereas the mutated version remains largely unaffected. The new data are presented in Figure 5.

We acknowledged that the Glu mutant would be a valuable addition to the CG-MD simulations represented in the old Figure 2. The new simulations were performed with different peptides in LDs containing or lacking PA. In line with our previous results with the full length proteins, we observed that both the wt and the (K167E/R171E/R175E) peptides could be embedded in the different LDs, but only the wt could directly interact with the PA molecules. These data are presented in the new Figure 5.

*Also, this reviewer would like to know more information about the specificity of the amphipathic helix interactions with PA. Would the interaction occur if the Arg residues were replaced with Lys or His, and if the Lys residue were replaced with Arg or His? Please report the conservation of all residues within the helix in vertebrate evolution.*

We have generated additional mutants to address this comment. The new data has been included in Figure 2. We have included the following text in the Results section to describe the results of the relevant substitutions:

“The absence of negative charges in the helix appeared as essential condition to permit TAG transference […] it can carry a positive charge depending on the pH and local environment which could explain the activity retained by this protein.”

We have amended the manuscript to include a figure (Figure 3—figure supplement 1) reporting the conservation of the amphipathic helix across vertebrate species. We have inserted the following text to describe our new analysis.

“K167 is 100% conserved across all vertebrate species examined. R171 was conserved across vertebrates including birds, snakes, lizards, crocodiles, turtles, marsupials, placental mammals, monotremes, although not in fish. R175 is also highly conserved with only birds, dolphin and Nile Tilapia (a fish) lacking this residue. Remarkably, an amphipathic helix is predicted in CIDEA of all the vertebrate species examined (Figure 3—figure supplement 1).”

Much of the data in the paper appears to be bands cut out of a western. Were all of these experiments performed on the same gel, or multiple gels? If the latter, please explain the controls for each experiment.

The CIDEA protein Western Blots after co-immunoprecipitation were prepared from separate transfections. In these experiments we are addressing whether there is a protein-protein interaction. Therefore, it is necessary to compare the immunoprecipitated protein with the input into the co-IP. Thus, we have included the protein input in the Western Blots, which shows the transfection was successful. A positive control of full-length wild type CIDEA confirmed that both the immunoprecipitation protocol and Western Blotting was successful.

*In Figure 4, it would be nice to show additional controls with other anionic phospholipids bound to beads – and examine competition with other anionic phospholipids. Also, testing the CIDEA mutant with the Glu mutations would be helpful. Similarly, Figure 4 and Figure 5 need additional controls.*

Our data clearly show binding of PA to CIDEA using lipid strips and PA-beads. Furthermore, in this assay we show interactions with other anionic phospholipids. We selected PA for further study and demonstrate the functional requirement of PA for the enlargement of LDs by CIDEA in both yeast and mammalian cells. We are not claiming that the CIDEA only binds PA, but demonstrating that binding occurs and this is required for the function of CIDEA. Although we agree that alternative lipid-beads or competition assays with alternative phospholipids could be done to test which other species could bind CIDEA, CIDEC and LIPIN-1γ, or if the Glu mutations selectively interfere with the binding of a subset of them, we consider that this information extends beyond the remit of this paper.

The specificity of the CIDEA interaction with PA in our assays with agarose-PA-beads was verified by the total lack of interaction with agarose-Protein A-beads. The strength of the assay was further validated with competition assays with soluble PA and PC. Once this interaction was confirmed, truncated CIDEA constructs were analysed to identify the regions involved in the interaction, for which we always included the control of the input protein in the same Western as the interaction with the PA beads.

Reviewer 2:

General comments:

*Figure 1–figure supplement 1 is a central figure and should be included in the main text.*

We have now included this figure as main Figure 2.

*Introduction and subsection “A cationic amphipathic helix in C-term drives LD targeting”: I would appreciate a more conservative approach in separating previously existing knowledge from new findings presented in this manuscript (e.g. of 3D structures or LD binding regions).*

The Introduction clearly outlines what are existing and new findings. The aforementioned subsection has been re-written and expanded to more clearly delineate our own analysis from previous findings (with references).

*Subheading “A cationic amphipathic helix in C-term drives LD targeting” and Figure 2: According to the hypothesis put forward by the authors, the short sequence 162-197 should target the N-terminal domain fused to the LD targeting helix to the LD. This should be shown experimentally and would strongly enhance the sole importance of this helix as targeting signal.*

We have undertaken the proposed experiment which included generation a plasmid to express a protein containing the N-terminal region fused to the region containing the amphipathic helix (HA-CIDEA-(1-117)-(163-180)). The results are shown and confirm that the N-terminal fragment alone did not target LDs whereas when fused to the amphipathic helix the resulting protein was competent in LD targeting and promoted LD clustering. Therefore, in addition to confirming the role of the amphipathic helix in LD targeting this result offers new evidence supporting the LD-LD docking activity of the N-term domain. These data are included in Figure 3 and described in the modified results section of the manuscript. Importantly, these data provide experimental evidence that our in silico analysis based on the 18-aa helix is a valid approach to model the interaction of CIDEA with the LD membranes.

*Subheading “A cationic amphipathic helix in C-term drives LD targeting” and Figure 2: CD completely lacks unit in x- and y-axis. Assuming that y-axis corresponds to MRE the negative peaks at 209nm appear to be extremely strong and beyond -35000. Are concentrations correct? The computed numbers of the CD spectrum should be related to the length of the measures peptide. What does 4% b-sheet mean in context of an 18-residue peptide?*

We have revisited the CD spectrum data. The concentration of the protein was measured by absorbance spectroscopy and the fit recalculated as we found the concentration lower than previously calculated based on the weight of the peptide alone. The re-analysed data has been re-plotted with units ∆ε (L mol^-1^ cm^-1^) and is included in Figure 3 with the fit estimates included in Figure 5—figure supplement 1. The peptides used in the CD study were 28 residues rather than 18 as stated in the Methods and thus also included non-helical flanking regions. The presence of b-sheet may indicate some degree of aggregation in the peptide solutions. Furthermore, peptides taken out of context of their full-length structures usually display a high degree of flexibility and thus display more random coil contributions than predicted based on sequence analysis.

In addition to re-plotting our CD analysis of the CIDEA amphipathic helix in detergent micelles, we also included new analysis of the wild type helix and helix with amino acids substitutions (F166R/V169R/L170R) in the presence or absence of liposomes. The addition of liposomes clearly induces the formation of helix (Figure 5). The axes are fully labelled in the graphs included in Figure 5 and are directly comparable to the detergent micelle spectrum in Figure 3. The detailed predictions of all secondary structure elements are provided in Figure 5—figure supplement 1. It is important to note that secondary structure predictions for membrane associated peptides and proteins are qualitative in nature and are included here primarily for ease of comparison and discussion of the data.

Paragraph two, subheading “A cationic amphipathic helix in C-term drives LD targeting” and Figure 2: Is the identified stretch of 18 amino acids embedded in a larger helix? Figure 2 gives the impression of a positive charge and negative charge at the N-and C-term of the helix respectively. Did the authors take care of the fact, that these are not charged NH3+ and COO- groups in context of the entire protein in their figures and MD calculations?

For the CG-MD, the 18 amino acids have been taken as is, without embedding them in a larger helix. This is because the circular dichroism experiments were conducted with a larger peptide that included flanking residues and there was no evidence that this would give rise to a larger helix. Also, the secondary structure predictions (Figure 3) do not indicate helix at the N-terminal side, and the C-terminal side only predicts a helical segment again after a break of 4 amino acids. Thus, given that there is no direct structural information that justifies the length of the helix, we only modelled the 18 amino acids that are strongly predicted to be helix. The reviewer is correct in noting the charges, and these have been omitted in the new simulations. Although the overall conclusions are not affected and the helix still binds to the lipid droplet, binding is now improved when PA was added to the system, in line with the experimental results.

*Paragraph two, subheading “A cationic amphipathic helix in C-term drives LD targeting”: The in-silico studies show a representation of the process that the helix diffuses into the micelles (assuming that the helix is already formed in the aqueous phase). However, since models were used I would be extremely careful making quantitative statements as "high affinity" or "quickly diffused" since they are not associated with quantitative numbers. These conclusions would require much more calculations as presented in this paper. Does an arbitrary 'control' helix e.g. from the N-terminal domain which is not involved in LD targeting not diffuse into the micelle or at a significantly lower pace?*

We have performed additional simulations with different helices to assess their relative affinity for the LD. In concordance with our cellular assays, we have confirmed that both the wild type and the K167E/R171E/R175E form a stable interaction with the LD and are embedded in the phospholipid monolayer. In contrast no LD binding was observed with the non-amphipathic mutants F166R/V169R/L170R. As suggested, we have included a control N-term helix, which did not interact with the LD. In concordance with the CD analysis in liposomes, the addition of 10% PA to the modelled LD rescues the impaired LD-targeting of F166R/V169R/170R, although it displays a superficial docking, confirming the importance of the helix amphipathic nature for its insertion in the LD membrane. The confirmation of the predicted behaviour of the different helices strongly supports the validity of this experimental approach to study the mechanism of action of CIDEA at the molecular level. We have not used quantitative statements such as “high affinity” or “quickly diffused” in our description of the new CG-MD data.

*Subheading “LD-LD docking is induced by the formation of CIDEA complexes”: The entire section puts forward very strong conclusions based on just one type of experiments. I suggest that other techniques could be used to test the protein-protein interaction.*

We have based our conclusions on two complementary approaches, namely Co-IP and confocal microscopy to determine LD-LD docking. The Co-IP is a reliable assay for determination of protein-protein interactions. Other assays such as GST pulldowns would not reveal additional mechanistic insight into the protein-protein interactions we have demonstrated. The dimerization of the N-term fragments of CIDE proteins had been previously characterized by NMR, X-ray crystallography, co-IP, and gel filtration assays, as referenced in the manuscript. The novelty of our approach is the combination of the co-IP data, to test the existence of interactions with the LD-LD docking analysis. This has provided the functional demonstration of the hypothesized role of CIDEA-CIDEA interactions in LD-LD docking, and has revealed that both the N-term and C-term dimerization sites can contribute to this process. To achieve this goal, our cell culture model was optimized to reduce the presence of spontaneous LD clusters to facilitate the study of CIDEA-induced LD clustering by the identification of LD-LD docking defective mutants. Numerous endogenous LD clusters are observed in undifferentiated 3T3-L1 cells, which have been previously used for functional studies on CIDE proteins. In contrast, our Hela cells contained few LDs in basal conditions, and LD formation was induced by BSA-oleate treatment 24 hours after transfection. The newly formed LDs were dispersed thorough the cytoplasm and were targeted by the different CIDEA constructs already present, which could promote their docking and enlargement during a short period of time before fixation (18 hours). This experimental setting largely prevented the LD-LD docking activity by endogenous LD proteins, allowing the disclosure of LD-LD docking defective CIDEA mutants.

Our new studies with HA-(1-117)-(163-180) offer additional evidence on the role of the N-term domain in LD-LD docking, showing the appearance of LD clusters when the soluble N-term fragment is directed to the LD membranes by the amphipathic helix.

*Paragraph three of the aforementioned subsection: I do not see the CoIP of CIDEA in Figure 3.*

This data has been included in the revised manuscript (now in Figure 4).

*Single point mutants in the N-terminal domain (R47E, E79Q/D80N) also show major phenotypic effects and should be discussed in more detail, especially in light that 3D structures with high sequence identify in these region are available. Interestingly, the LD pattern of the variants R47E (Figure 3) compared to R47A and R47Q (Figure 1–figure supplement 1) is drastically different and should be discussed in more detail.*

We have discussed in more detail the effects of amino acid substitutions in the context of the CIDE-N domain in light of its 3D structure and in relation to CIDEC. The following passage was included in the manuscript:

“The lack of co-IP between the N-term fragment and the full length CIDEA could be due to conformational and positional factors […] the N-term domain of CIDEA can contribute to LD-LD docking by forming complexes with its counterparts on the adjacent LD.”

Reviewer 3:

Suggested additional experiments:

*1) Prove interactions of AH, native sequence and lack of interactions of mutated control sequences, with TAG emulsions. The CD data of peptide in solvent is of no help here. If an experiment showing increased helicity with lipid binding can be devised this will be helpful also.*

We thank the reviewer for this suggestion. We have undertaken new CD analysis of CIDEA peptides in aqueous buffer in the absence and presence of liposomes. These data show that helicity of the CIDEA AH is enhanced in the presence of liposomes. Significantly, we show that amino acid substitutions that eliminate the amphipathic nature of the helix reduce helical folding even in the presence of liposomes (DLPC).

2) Show that interactions with TAG emulsions increase (or otherwise modified) with addition of PA to surface monolayers.

We were indeed able to show that with CD analysis while DLPC liposomes were unable to promote folding of the CIDEA helix with amino acids substitutions F166R/V169R/L170R (designed to eliminate the amphipathic nature due to additional charge of R compared to the hydrophobic residues in the WT), DLPA:DLPC (1:9) liposomes were able to facilitate helical folding.

Using CG-MD simulations to model the molecular dynamics of the helix-LD interaction we have observed that PA is not required for LD targeting, confirming our previous observations separating PA-binding from LD-targeting. However we have observed a faster interaction of the wt helix with LDs containing PA. Remarkably we have observed the direct interaction of PA with the polar residues of the wt helix, whereas the inactive K167E/R171E/R175E avoids the PA molecules. This has offered further evidence to support our molecular model for the activity of CIDEA.